Resource

# Full-scale scaffold model of the human hippocampus CA1 area

Daniela Gandolfi [1,2,9] ✉, Jonathan Mapelli [1,3,9] ✉, Sergio M. G. Solinas[4,5], Paul Triebkorn [6], Egidio D'Angelo[2,7], Viktor Jirsa[6] & Michele Migliore [8] ✉

The increasing availability of quantitative data on the human brain is opening new avenues to study neural function and dysfunction, thus bringing us closer and closer to the implementation of digital twin applications for personalized medicine. Here we provide a resource to the neuroscience community: a computational method to generate full-scale scaffold model of human brain regions starting from microscopy images. We have benchmarked the method to reconstruct the CA1 region of a right human hippocampus, which accounts for about half of the entire right hippocampal formation. Together with 3D soma positioning we provide a connectivity matrix generated using a morpho-anatomical connection strategy based on axonal and dendritic probability density functions accounting for morphological properties of hippocampal neurons. The data and algorithms are supplied in a ready-to-use format, suited to implement computational models at different scales and detail.

In recent years, research on computational brain models has increased rapidly, leading to a large set of data-driven models[1]. Large-scale implementations of brain circuits at single-cell resolution have been proven to be instrumental for a better understanding of brain functions and could become a disruptive technology to investigate pathological conditions and discover new pharmacological treatments. Despite extensive efforts and remarkable investments (for example, Human Brain Project[2], Human Connectome project[3], The Virtual Brain project[4], Human Neocortical Solver[5], Openworm[6] and Open Source Brain[7]), the lack of critical data (for example, morphology, electrophysiology, synaptic properties and connectivity) on human neurons and circuits suggesting how and to what extent they differ from other species still substantially hinders our understanding of the specific mechanisms underlying brain functions in humans.

Cellular data on human brain are sparse[8] and mostly limited to a few neocortical regions. Technological and methodological limitations have prevented the possibility to collect enough experimental data on human brain at the cellular level[9–11]. The simulation of brain activity at cellular resolution with large-scale model has been obtained for different animal species and entire rodent brain areas[12–15]. Notably, novel co-simulation technologies, in which regions of interest are modelled at high cellular resolution and others using dimension reduction techniques[16], enable mixed modes of operation. Such modes have been developed for co-simulations of the NEST simulator[17] and The Virtual Brain[18].

With respect to connectivity, despite several technological advancements[19–21], the most widely adopted method to analyse entire human brain samples is light microscopy of silver- or Nissl-stained samples. Unfortunately, the non-specificity of both labels[22] limits the spatial resolution to neuronal size. Beside data collection, different strategies to connect neuronal networks ranging from randomized connectivity to 'touch detection' algorithm[23,24] have been proposed. However, the computational load and the scarce availability of data required to generate realistic models must be considered when connecting

[1]Department of Biomedical, Metabolic and Neural Sciences, University of Modena and Reggio Emilia, Modena, Italy. [2]Department of Brain and Behavioral Sciences, University of Pavia, Pavia, Italy. [3]Center for Neuroscience and Neurotechnology, University of Modena and Reggio Emilia, Modena, Italy. [4]Department of Biomedical Science, University of Sassari, Sassari, Italy. [5]Institute of Neuroinformatics, University of Zurich and ETH Zurich, Zurich, Switzerland. [6]Institut de Neurosciences des Systèmes, Aix-Marseille University, Marseille, France. [7]IRCCS Mondino Foundation, Pavia, Italy. [8]Institute of Biophysics, National Research Council, Palermo, Italy. [9]These authors contributed equally: Daniela Gandolfi, Jonathan Mapelli. ✉e-mail: daniela.gandolfi@unimore.it; jonathan.mapelli@unimore.it; michele.migliore@cnr.it

millions of neurons. Variants of 'touch detection' algorithms, are in fact based on morphologies derived from experimental data. Alternatively, probability distributions of axonal and dendritic volumes have also been proposed[25]. These methods estimate the connectivity among neurons through isotropic or distance-dependent criteria[26]. Moreover, neuronal connectivity can be derived from the conversion of axons and dendrites into grids of voxels generating density fields of neurites whose intersections determine the probability of contact[27]. However, procedures customizing morphological orientation according to specific constraints are particularly complicated in human brain circuits, due to the complexity of anatomical organization[28].

In this Resource, we provide a computational method to generate the 3D positions and full connectivity of brain regions starting from microscopy images. We have benchmarked the method to the Cornus Ammonis-1 (CA1) region of a right human hippocampus by creating the 3D positioning of all pyramidal cells (PCs) and interneurons as well as the full network connectivity.

## Results

### Model overview

The method pipeline is represented in the flowchart shown in Extended Data Fig. 1. The workflow is divided into sequential blocks: (1) neuronal placement, (2) neuronal morphology, (3) network connectivity and (4) network simulation.

### Neuronal placement

Neuronal placement was performed by analysing a dataset of human brain images (Bigbrain[29]; Fig. 1a) previously labelled[30] for the subregions of the hippocampal formation (CA1,2,3,4, dentate gyrus (DG) and subiculum (Sub)). Labelled images were segmented and employed to generate the surface of hippocampal regions adopted as external anatomical landmarks. The calculation of hippocampal subregions confirmed the estimates that the volume of CA1 (547.1 mm³) is about half of the entire hippocampal formation (Sub 289 mm³, CA2 41.6 mm³, CA3 55.6 mm³, CA4 110 mm³ and DG 110 mm³).

According to the labels obtained in the first analysis, stacked images were cropped to isolate the regions of interest (Fig. 1b). Binary images were then generated through dynamic thresholding on hippocampal stained areas (Fig. 1c,d and Methods) and were then converted into 3D coordinates according to the $x,y,z$ resolution (Fig. 1e). The image-processing procedure identified about 18 million cells. However, given the non-cell type specificity of the staining method, voxels contained a mixture of glial and neuronal cells to be in turn further differentiated into PCs and interneurons.

The hippocampal network is a conserved brain structure, and despite the marked surface gyrifications and complex folding of the human CA1 (Fig. 1 and Discussion), PCs show similar morphological characteristics among mammals from rodents to human. Given these premises, we have assumed that human CA1 neurons could be divided in excitatory and inhibitory with a further subdivision of GABAergic interneurons like the one encountered in rodents. From the 18 million cells, 5.28 million were randomly selected to represent PCs[31,32] and interneurons[33]. The number comes from the estimate of PCs reported in ref. [32] (4,836,111 cells) and subsequently rounded to the closest integer. The GABAergic interneurons have been classified by adopting a 10% ratio between excitatory and inhibitory neurons: a proportion well conserved among species (from 5% (ref. [34]) to more than 20% (ref. [35])) and brain regions, including hippocampal circuits[36,37]. We have adopted an intermediate value of 10%, in agreement with the range of interneurons representativeness reported in refs. [36,37] and in experimental databases such as https://bbp.epfl.ch/nexus/cell-atlas/.

The neurons were therefore subdivided into two classes, labelling 90% (4.8 M) of the overall population as PCs (pink circles in Fig. 2) and the remaining 10% (0.48 M) as interneurons. According to the terminology adopted for rodents[38], interneurons were further grouped into seven classes based on their morphological features and location within hippocampal layers (Fig. 2a,b and Methods).

It has been observed that PCs are distributed according to a bidirectional gradient along the medio-lateral[39,40] and antero-posterior axes[41]. We have calculated the neuronal density distribution of the model along the dorso-ventral (transversal slices, Fig. 3a) and medio-lateral directions (Methods). In contrast with rodents (compare Figs. 2 and 3 in ref. [39] and Fig. 2 in ref. [40]), where PCs are mainly aligned within a thin stratum pyramidale (SP) layer, the human SP is much thicker (~1 mm) and cells are anisotropically distributed, with a preferential accumulation in the proximity of the stratum radiatum (SR) (Fig. 3b). The analysis performed in transversal slices showed a marked increase in the cell density going from the stratum oriens (SO) to SR (Fig. 3c, +289 ± 71%; Methods). The density profile was also estimated in the antero-posterior direction by sampling voxels (white circles in Fig. 3d; Methods) along that axis. The average density profile in the antero-posterior (A-P) axis showed a marked increase (+127 ± 21.6% mean value of the density in the first four voxels in anterior part versus the last four voxels in the posterior part; Fig. 3e) in agreement with previous suggestions[30,41].

### Neuronal morphology

Neuronal morphology is performed by translating the shape of axons and dendrites of PCs and GABAergic interneurons into geometrical probability distributions that were parameterized according to the values reported in Supplementary Table 1. Cells were oriented according to their relative positions with respect to anatomical landmarks. More specifically, the human CA1 PCs have basal and apical dendrites whose shapes can be approximated to cones oriented towards the SR and SO. This geometrical feature, which closely resembles the one encountered in mouse PCs[25], suggested the adoption of conical shapes for both apical and basal dendrites of human PCs. Furthermore, according to literature[39], dendritic extension of human PCs is about 150% of mouse dendritic size (see Fig. 6 in ref. [39]). The estimated values of mouse pyramidal dendrites[25] were therefore rescaled by a factor of 1.5 (Methods). Conversely, PC axons could not be approximated to regular shapes. Differently from rodents in fact, axonal pathways of PCs had to be adapted to the marked sulci and gyrification characterizing the human hippocampal surface (Fig. 4). Given the lack of data, we have assumed a similarity with rodents where pyramidal axons preferentially branch towards the subiculum (Fig. 4b) with a poor backpropagation to the CA3 (see Fig. 4b inset and the MouseLight database) and a limited longitudinal spread. The axonal probability density functions were generated by calculating transversal planes for every PC (Fig. 4b and Methods). The transversal planes were used to calculate the pathway of the axon extending towards the SO and running towards the subiculum (Fig. 4c and Methods). Moreover, the axonal probability density functions have been generated by creating a tubular structure with a 300 μm diameter running in parallel with the CA1 from the soma positioning towards the subiculum and with a limited back-projection (~150 μm) to the CA2 (Methods and Fig. 4c). The result of the overall procedure is illustrated in Fig. 4d,e showing examples of axonal and dendritic probability distributions of PCs.

The full network was constructed by adding GABAergic neurons whose geometries were designed according to the morphological properties derived from rodents[25] (Methods, Supplementary Fig. 1 and Supplementary Table 1). We have assumed that the major inhibitory classes that are present in rodents could also exist in human. By analysing the morphological features of experimentally reconstructed mouse CA1 interneurons, seven morphological classes representing the heterogeneity of GABAergic CA1 cells[25] were identified. Each of these classes was composed of a combination of ellipsoids and cones (Methods), and the parameters derived from the analysis of mouse morphologies[25] were corrected by a factor of 1.5. To increase the variability, values adopted to parameterize neuronal geometries

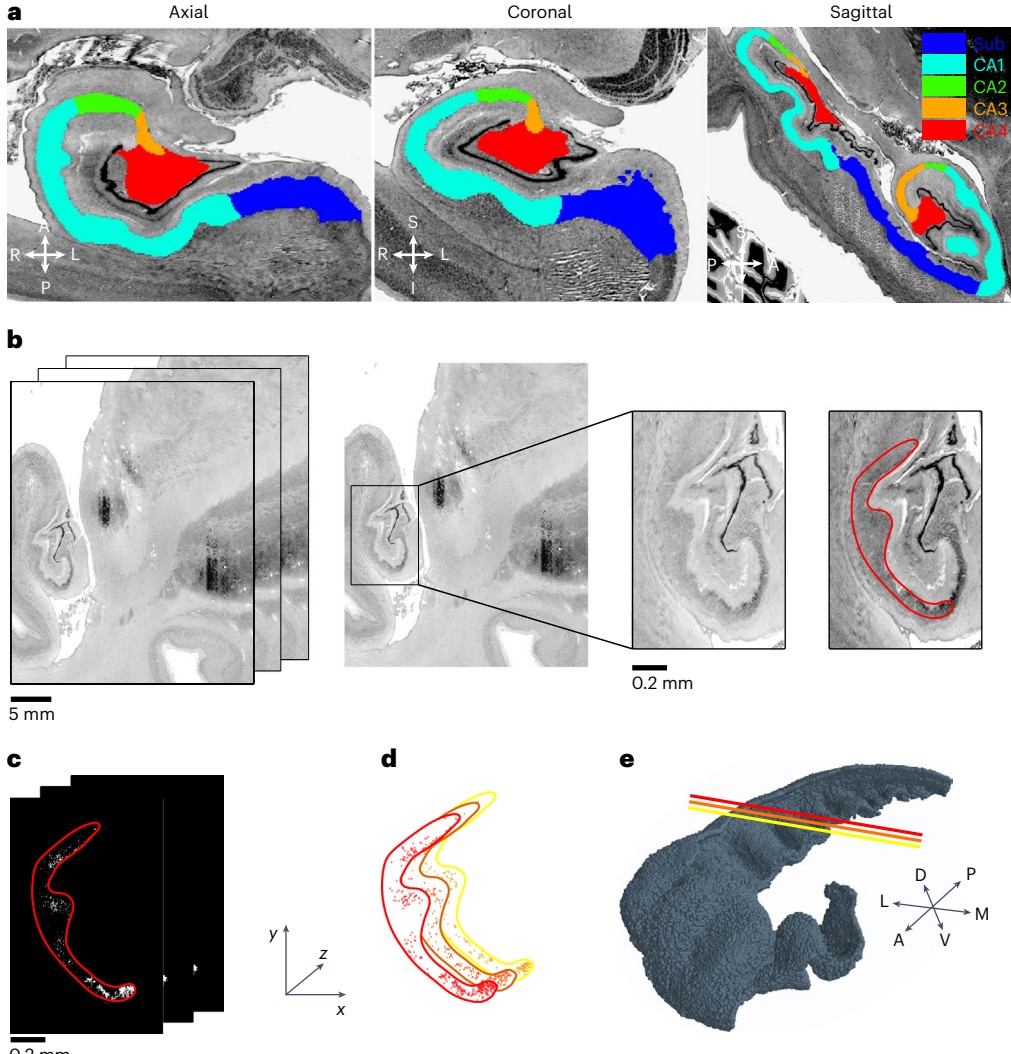

**Fig. 1 | Single image analysis. a**, Cell body-stained histological sections of a right hippocampus from the BigBrain[29] database at 20 μm voxel resolution. Overlay of the manual segmentation of the hippocampal formation from ref. [30]. **b**, Stack images of silver-stained coronal sections from a 65-year-old male human brain ($x,y,z$ resolution $20 \times 20 \times 20$ μm³; BigBrain[29]). Centre: stack images were cropped (black box) to isolate hippocampal structures. Right: cropped images were automatically adjusted through an image-processing algorithm highlighting cell bodies. **c**, Binarized images resulting from image segmentation and corresponding to cell body positions (white spots) within the CA1 subregion (red contour) isolated from the background. **d**, $x,y$ coordinates of cell bodies (red spots) are assigned on the basis of the pixel grid, while $z$ coordinates correspond to stack level (red contour corresponds to red contour in **c**). **e**. 3D cell body distribution of a complete right CA1 hippocampus (coloured lines represent coronal planes shown in **d**).

were randomly chosen within normal distributions of dendritic and axonal sizes (Supplementary Table 1).

**Network connectivity**

The connection pairs were obtained by intersecting the convex hull of every presynaptic neuron with dendritic points of postsynaptic neurons. The connectivity matrix was created by intersecting axonal and dendritic probability density functions of the different neuronal classes. To reduce computational time, the connectivity of each neuron was calculated only in a limited region determined by the overlapping of axonal and dendritic bounding boxes (Extended Data Fig. 1). Independently from the number of dendritic points included in the axon, every pair of intersecting neurons was included in the initial connectivity matrix.

The resulting human CA1 connectome generated about 40 billion of connected pairs, in good agreement with the estimation based on the rodent connection probability (www.hippocampome.org (ref. [42]); Supplementary Table 2), once rescaled to human neuronal numerosity.

The model network architecture was validated by evaluating the probability density of converging inputs and diverging outputs, also called indegree and outdegree. It has been reported[43] that the outdegree and indegree distributions of experimental neuronal networks in different brain regions share similar features. The initial evaluation was performed exclusively for the excitatory network of PCs (Fig. 5a,b, pink area). The indegree and outdegree curves exhibited shapes consistent with experimental data (see Figs. 1 and 6 in ref. [43]). A similar profile was conserved including inhibitory connections albeit a shift to higher values for the peak of the distributions (from 1,100 to 1,600 for the outdegree and from 2,400 to 3,300 for indegree). The similarity between the two distributions was estimated through the Kullback–Leibler (KL) divergence method. A difference between the two conditions (0.029 KL score for outdegree and 0.033 KL score for indegree), probably due to a wider probability density function of interneurons (Supplementary Fig. 1), was observed. In both cases, the connection length distribution (Fig. 5c) had shapes similar to that observed experimentally in rodents and in other brain areas[43]. The inhibitory connections shifted the peak

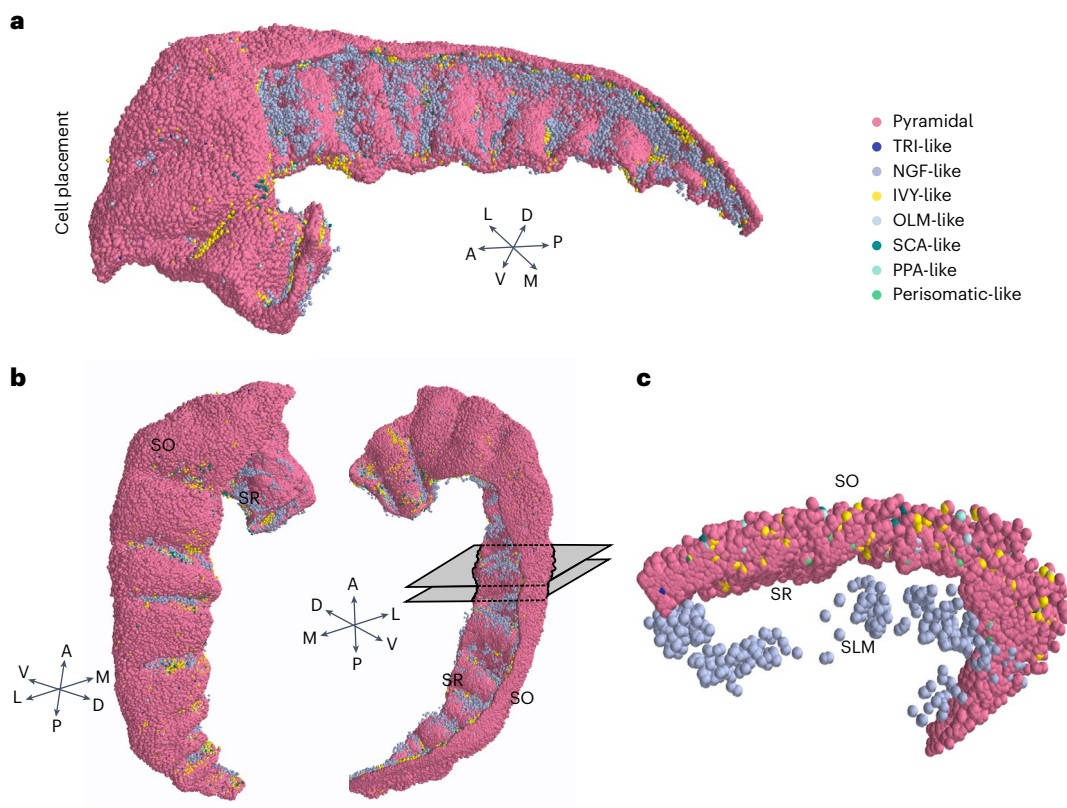

**Fig. 2 | Neuronal soma positioning. a**, 3D positioning of the excitatory (PCs, pink) and inhibitory neurons. Interneurons are divided into seven classes according to positioning and morphological features. **b**, Reoriented 3D neuronal positioning shown in **a** to highlight interneurons distribution. **c**, Two-millimetre transversal slice of 3D positioning obtained from sectioning CA1 between the two grey-shaded planes shown in **b**, right. Note the NGF-like neurons (grey spots) in the lower part corresponding to SR and SLM and IVY-like neurons (yellow spots) scattered within the pyramidal layer (SP) derived from the sampling procedure adopted to place interneurons and described in Methods.

of the distribution to higher values (from 1,100 μm to 1,900 μm) while the subtracted curves show that larger connection lengths are present in the purely excitatory network. The connection strategy was further validated by analysing indegrees and outdegrees in a randomly connected network. The profiles distribution obtained from random connectivity showed a Gaussian shape with peak values around 5,400 connections with a narrowed half-width (150 connections, see Supplementary Fig. 2) for both indegree and outdegree in contrast with the expected profiles obtained from experimental observations[43] (0.7 KL score). Conversely, the proposed approach considers a more realistic cell distribution, maintaining the natural anatomical layout and the peculiar properties of the hippocampal CA1 neurons and interneurons organization. The proposed model allowed one to observe hub neurons (Fig. 5), highly connected elements playing a key role in hippocampal computation[44] whose emergence was prevented by random connectivity (Supplementary Fig. 2). The dependence of the network connectivity from parameterization was evaluated by changing axonal and dendritic parameters, and results are shown in Supplementary Fig. 3. The KL estimates revealed that the shape of the probability densities was not affected in all the configurations (axons and dendrites were halved, doubled or left unchanged; KL average score 0.012 with a minimum of 0.0016 and a maximum of 0.042; less than 1/10 compared with the network generated with random connectivity).

### Network simulation

A model network, using simple integrate and fire neurons and Tsodyks–Markram synapses with default parameters[45], was implemented with the exclusive intention to have a demo model to test running times and circuit integrity. The computational effort required to run a full-scale simulation was tested using the excitatory network (connection matrix stored in HDF5 format with gzip compressed dataset, file size approximately 86 Gb) implemented in NEST simulator (https://www.nest-simulator.org). For the purpose of this work, neurons were implemented as standard Hill and Tononi point neurons, available in the NEST distribution ('HT_neuron'). The neurons were connected with 'Tsodyks–Markram' synapses, also available in NEST ('Tsodyks_synapse'). Hippocampal activity was stimulated by activating with a single pulse a group of 6,706 PCs distributed in a spherical volume of 500 μm radius. Results of a simulation performed on a purely excitatory network are shown in Fig. 6 (for the full video, see Supplementary Information), where it can be evidenced that, in response to a single stimulus delivered to a confined population of PCs, the activity starts spreading transversally (CA2–subiculum) to be further diffused longitudinally (Fig. 6). The same stimulation protocol has also been arranged for a simulation with the complete network (Supplementary Figs. 4 and 5) showing that inhibitory circuits prevent the signal spread in the longitudinal direction. Full-scale network test simulations were carried out on the Piz-Daint Cray XC40 supercomputer available at the Swiss National Supercomputer Center (CSCS, ETH Zurich), composed of 1,813 nodes each featuring two Intel Xeon E5-2695 v4 @ 2.10 GHz (2 × 18 cores, 64 or 128 GB random-access memory). The large number of synaptic connections composing the CA1 network, about 40 billion, required a special setup procedure. To minimize the memory requirements, excitatory connections were loaded and instantiated in chunks of 166 million taking a total of about 10,000 s. The inhibitory connections were loaded also

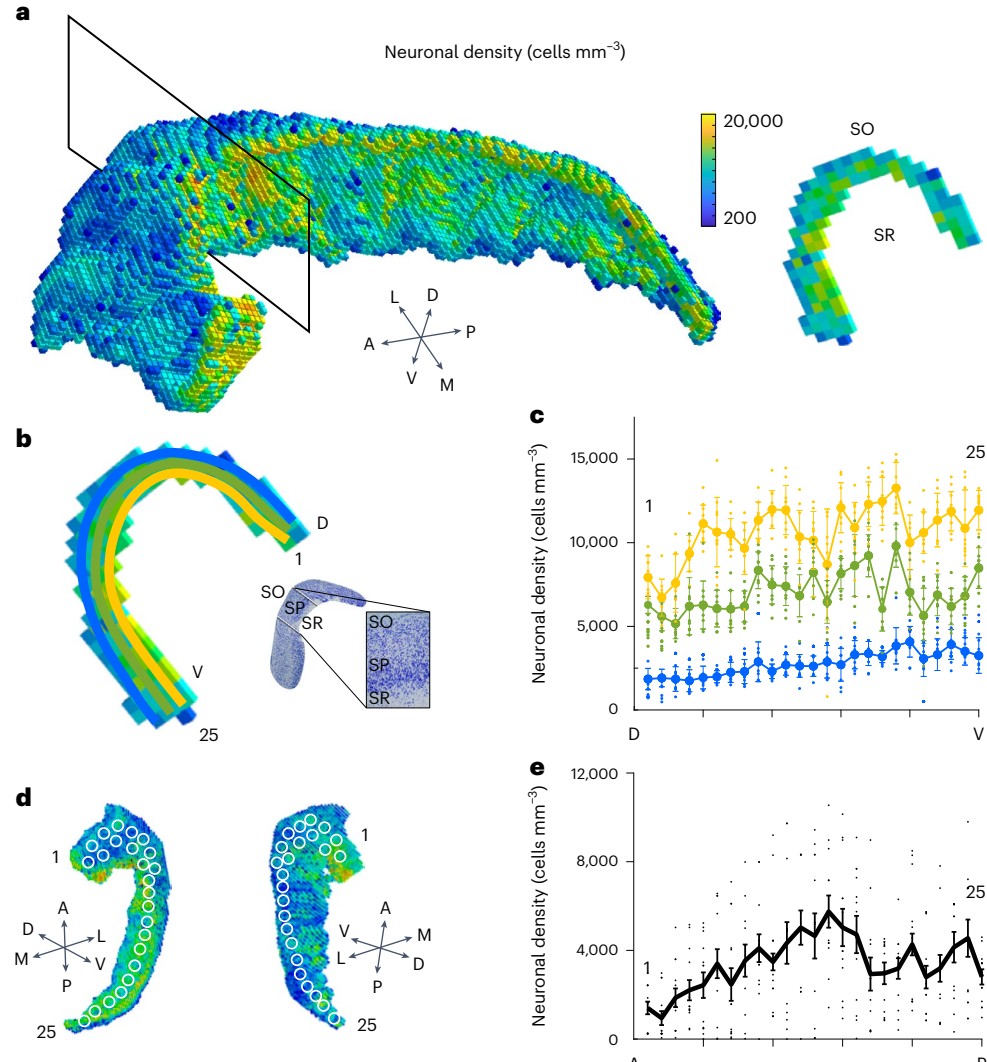

**Fig. 3 | Analysis of neuronal density. a**, Left: 3D voxelization of neuronal placement: each voxel ($300 \times 300 \times 300$ µm³) is coloured according to its neuronal density value, from low (blue) to high (yellow) density. Right: transversal slice of a single layer of voxels obtained from the slicing plane on the left (black rectangle). Note the higher density in the SP and SR compared with SO. **b**. Density profile along the medio-lateral axis. Coloured spots have been obtained by calculating density values for 25 voxels in three stripes of the SP running from the dorsal (D) to the ventral (V) side of the CA1 in analogy with the density gradient within SP shown in the images in the inset (adapted from ref. [53]). Blue dots (lateral, SO side), green dots (middle), yellow dots (medial, SR side). **c**, The analysis has been repeated for ten transversal slices (+289 ± 71%; mean ± s.e.m.). **d**. 3D voxelization of neuronal placement with 25 positions (white circles) employed to calculate the density distribution. **e**, Density profile along the antero-posterior direction obtained sampling 25 different positions (white circles in **d**). Black line represents the average of 13 different sampling voxels ($1,000 \times 1,000 \times 1,000$ µm³) for each position (+127 ± 21.6; mean ± sem).

from hdf5 files, and their creation took about 4,000 s. Moreover, the maximum number of connections that could be managed by NEST on each computing node cannot exceed 134,217,727, to respect limitations of local indexing of instantiated elements. To meet this limit, the number of central processing units (CPUs) (tasks) cannot be less than 250. To meet memory requirements, we used a total of 160 nodes and allocated at most three tasks on each computing node, using a total of 480 processes to set up the network. Since each process was instructed to use five threads to carry out the simulation, the actual number of processors used to simulate the full network was 2,400, the available amount of random-access memory was 19.2 TB out of which 7.2 TB was used by NEST.

Disregarding the network setup time, 200 ms of simulated time and data saving in the native text file format of NEST (.dat files) required 162 s of CPU time.

## Discussion

One of the main assumptions of this work is that some of the morphological and anatomical properties of hippocampal formations can be conserved during phylogenesis. We have assumed that features extracted from experimental observations in rodents could be translated into human structures. Our scaffold model considers the known differences at both macro- and microscopic level between human and rodent hippocampus, like the marked surface gyrification or a thicker SP with sparse PCs. These structural features might have functional consequences on the activity of the whole network, which remains to be tested through a suitable implementation of human single-cell and synapse-computational models. This is crucial, since it has been shown that cortical gyrifications correlate with information-processing capacity[46] while their alterations could lead to cognitive impairment[47] and are correlated with neurodegenerative diseases[48]. The data extracted

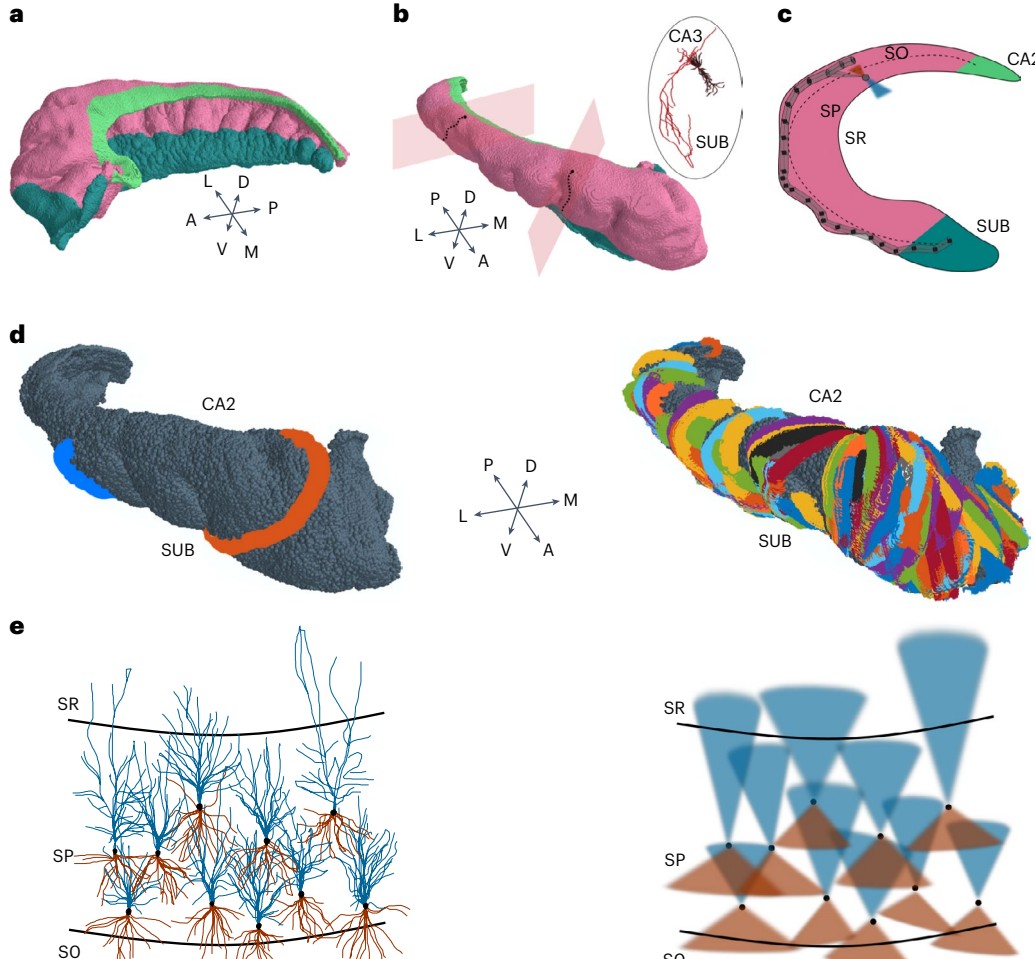

**Fig. 4 | Generation of axonal and dendritic probability density functions.**
**a**, 3D meshes of CA1 (pink), CA2 (light green) and subiculum (dark green) generated by stacking the segmentation boundaries of hippocampal subregions. **b**, Left: orientation planes were generated for each PC (black spots) according to the relative distances of PCs from CA2 and subiculum (Methods). Axons (black dotted line) are created in the orientation plane and project towards subiculum adapting to CA1 surface gyrification (Methods). The inset shows the reconstruction from experimental data of the entire morphology of a murine CA1 PC (Janelia Research Campus http://mouselight.janelia.org/). Note the directionality of PC axons from CA3 side to subiculum. **c**, Schematic representation of the axon modelling procedure. The points of the CA1 surface laying on the orientation plane are connected through a spline line defining a tubular volume (150 μm radius). **d**. Left: example of modelled PC axons (orange and blue thick lines) running in the SO from PC somas placement towards subiculum. Right: 100 randomly selected PC axons running in the external part of the SP from PC placement towards subiculum. **e**. Left: realistic morphology of PCs (basal dendrites in brown and apical in blue, adapted from ref. [53]), oriented within a transversal CA1 hippocampal slice. Right: the probability density functions are represented as two cones with colour code respecting the realistic morphology.

to implement models of the hippocampus at cellular resolution will be used to run co-simulations, a framework enabling the use of two simulators operating at different scales to run simulations of full brain networks (Supplementary Fig. 6). A further embedding in a large-scale network using co-simulation will shed light on the relevance of these structural variances at the large-scale network level (Supplementary Fig. 6). Using co-simulation would enable the modelling of the impact of the pathomechanisms on the emergence of temporal lobe epilepsy, which arises at the microscale level[49], on the full brain network. We expect that those models will lead to new mechanistic hypotheses, as well as to an improved prediction of patient-specific seizure dynamics.

We have assumed that both excitatory and inhibitory neurons had to be distributed according to the mapping generated by image analysis procedures. In particular, inhibitory interneurons have been placed in analogy with general morpho-anatomical features observed in rodents. We have adopted an anisotropic distribution for GABAergic interneurons because: (1) recent findings showed that human GABAergic hippocampus interneurons exhibit strong similarities with GABAergic mouse cell types and could be clustered into seven classes depending on expression patterns of marker genes, and global transcriptional similarity[48], and (2) the distribution of GABAergic interneurons in single human cortical columns, which maintains a fairly constant ratio of about 10% with respect to glutamatergic neurons throughout layers, exhibits a marked asymmetry and layer specificity for different subtypes[50]. Similarly to rodents, inhibitory neurons in human isocortical columns are clustered within distinct preferential layers providing functional segregation of inhibitory patterns. It is therefore not plausible to obtain a diffuse inhibitory action by randomly and isotropically arranging neuronal soma throughout the simulated volume.

These observations support our hypothesis of an anisotropic distribution of GABAergic subtypes that can presumably resemble that observed in rodent hippocampus, albeit molecular similarities and cortical distribution in humans do not necessarily imply a quantitative match of morphological and topological features between humans and rodents. Furthermore, the use of a unique inhibitory class distributed isotropically in the CA1 volume would introduce

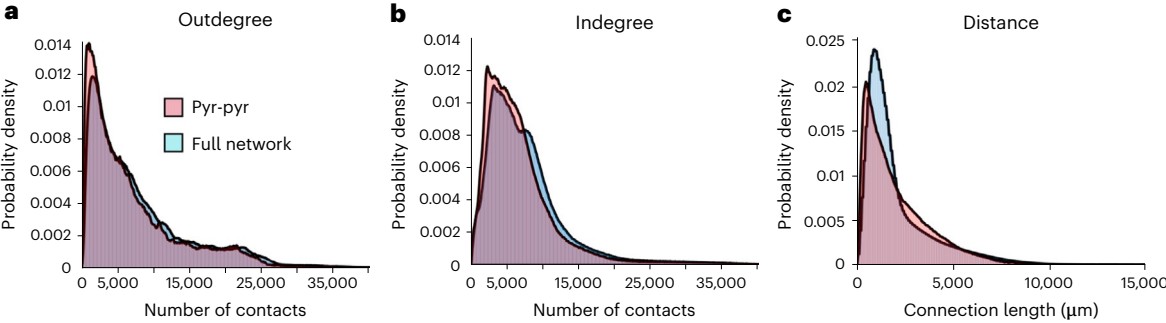

**Fig. 5 | Indegree and outdegree. a,** Probability density of the number of neurons contacted by every neuron (outdegree). The integration of inhibitory synapses in the network (blue histogram) shifts the peak of the curve (from 1,100 contact with 0.0148 probability density to 1,600 contacts with 0.0126 probability density) and induces an increase of the probability density at larger numbers of contacts (note the prevalence of a blue profile at larger number of contacts). **b,** Probability density of the number of contacts received by every neuron (indegree). The degree distribution shows a peak of incoming input of 2,400 units. Including inhibitory synapses in the network, the curve increases and the peak slightly shifts (from 2,400 to 3,300). **c,** Probability density of connection lengths. Note the shape closely resembling results obtained from rodents[42,43]. In the presence of inhibitory synapses, connection length distribution shifted to right (from 1,100 to 1,900 μm).

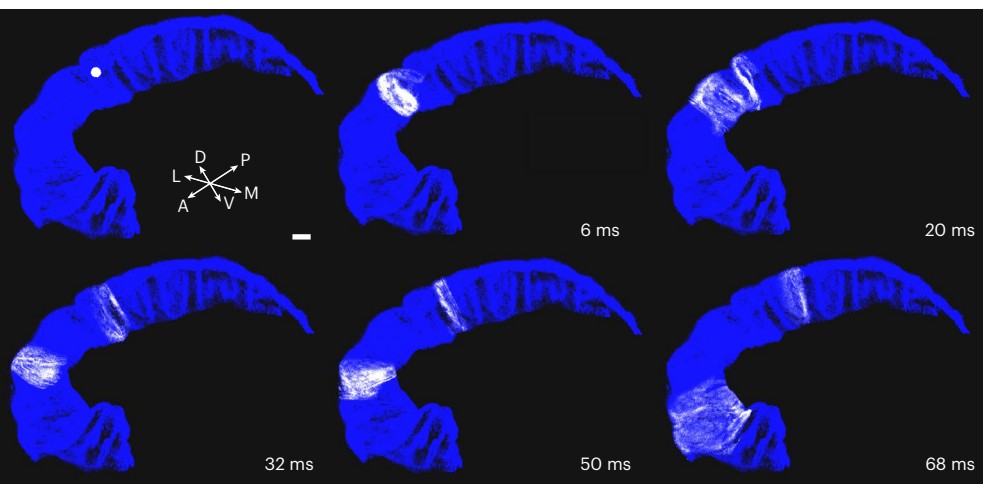

**Fig. 6 | Full-scale network simulation.** Snapshots from a demo movie (Supplementary Movie 1) illustrating a simulation of a purely excitatory network in which the activity was evoked by a single pulse stimulation delivered to about 6,500 PCs in a 500 μm radius sphere near the CA2 region. Note that activity initially propagates in the transversal (medio-lateral) direction to subsequently spread longitudinally (antero-posterior). Scale bar, 2 mm. Neuronal firing is coded by single neurons turning white (spiking) from blue (silent). Images generated with ViSimpl (https://vg-lab.es/visimpl/).

an artefact in the choice of the most probable shape to be assigned to this neuronal model.

It is indeed well consolidated that human PCs are distributed in a thicker SP[51] compared with rodents and preserve morphologies with some differences in dimensions. This spatial segregation based on the radial axis could represent a general principle that, in rodents, is implemented by the deep and superficial PCs that act increasing the capacity to compute and to perform different tasks in parallel[52]. Accordingly, the distribution of PCs in our scaffold, by following a medio-lateral gradient running from deep to superficial layers could instantiate parallelized computation. Furthermore, this organization implies that (1) pyramidal apical and basal dendrites are sparsely organized within the SP, and (2) inhibitory neurons and neurites should be reorganized to quench the activity of excitatory neurons in a way that it is largely unknown. In this work we hypothesize that human interneurons, as well as PCs, maintain their morphology features adapting the size and sparsely populating the different layers in compliance with an inhomogeneous cellular density distribution. Starting from the experimental data reported in ref. [53] (see Fig. 7 in ref. [53] on the primate CA1 cellular layer organization), only deep PCs project their basal dendrites to the SO and only superficial PCs project their apical dendrites to the SR and stratum lacunosum-moleculalris (SLM). Thus, we hypothesize that dedicated classes of interneurons populating the SO or the SR will target more specifically pyramidal apical or basal tufts, while what we called perisomatic interneurons inhibit the PC population acting generically on perisomatic dendrites (basal or apical). We can conclude that the choice of modelling seven different classes of interneurons provides the basis for translating knew knowledge into the model, as it will be available, starting from the plausible assumption that human interneurons should have different morphologies to provide dedicated inhibition.

The sparseness of data at the cellular resolution in human brain research has compelled the generalization of the model through arbitrary assumptions that were instead largely inspired by rodents cyto-architecture. Furthermore, the parameterization of the model with a few features for each neuronal class could expand the possible network configurations. Our assumption, beside the choice of specific geometrical shape mimicking neuronal morphologies, was dictated by the experimental observations that PCs show a scale factor of 1.5 from mouse to human. The same value has been translated

to morphological parameters of interneurons. It should be noted that changing network configuration either by randomizing connectivity or by changing morphological parameters has a marked effect mainly on the size of the CA1 connectome, which becomes unrealistic when the network is randomized. Halving or doubling PC axons and dendrites impacts the total number of connections while poorly conditioning the indegree and outdegrees and the connection lengths probability distribution.

The proposed model together with the computational method is a starting point for generating more sophisticated models incorporating the functional characteristics that are required to simulate the entire hippocampus. As a starting point the model suffers for some limitations: (1) the image analysis has been performed on images obtained with a low resolution and low specificity method; (2) the choice of the parameters is based on the assumption of similarity between mouse and human, which is currently unknown; (3) the use of geometrical probability density functions rather than realistic morphologies is a strong generalization; (4) the absence of the CA1 layers as automatic internal landmarks required to adopt a division of CA1 surface in a deep and a superficial side. The generation of a model describing a complicated biological system like the human brain requires assumptions and reductions that progressively scale up with the complexity of the system. The current model could be therefore expanded and refined by (1) using updated morphological data in particular of human inhibitory neurons; (2) analysing human samples to obtain updated values for human morphologies; (3) refining probability density functions to obtain a more sophisticated morphological representation of neuronal classes or, alternatively, creating realistic synthetic neurons; (4) labelling CA1 layers to be adopted as additional internal landmarks. Finally, among the differences between the rodent and human hippocampus, the anatomical organization of the human CA1 closely resembles cortical structures where gyrification allows to expand the grey matter area maintaining the volumetric size. From a computational perspective, this structural organization, whose alterations have been correlated to the occurrence of neurological disorders such as autism[54], remarkably expands the complexity of the tissue and hampers any chance to implement models based on randomized connectivity strategies. Conversely, the proposed connectivity method and the generated scaffold model take into account the geometric convolution of CA1 by developing axons following CA1 surface and potentially preserving the functional consequences of gyrification such as the emergence of cognitive functions[55]. Moreover, the proposed model reproduces the antero-posterior gradient of neuronal density that has been experimentally observed[53] and theoretically predicted[22]. This aspect is paramount, since external connections impinging onto CA1 are non-homogeneously distributed and follow an antero-posterior gradient[52] that recalls the anisotropy in cellular distribution.

In conclusion, this scaffold model of a CA1 human hippocampus, if properly equipped with realistic models of neurons and synapses, will promote the development of a full model of the human hippocampus allowing the investigation of its function and providing a valuable digital tool for the development of better treatments for neurological diseases.

## Methods

The automatic detection of 2D cell body was designed to analyse single-channel marker microscopy images, labelled for the identification of the hippocampal subregions. The algorithm, written in MATLAB (v 2019b; The Mathworks Inc), was conceived to be applied to silver-stained sections. The staining procedure, which darkens cell bodies in an unspecific way, did not allow us to assign a specific neuronal labelling to a given population.

A two-step procedure has been implemented to analyse labelled and raw images: image labelling and image segmentation.

### Image labelling

The BigBrain imaging dataset[29] is a high-resolution microscopic full human brain scan at a resolution of 20 μm and labelled for the different hippocampal subregions. The manual labelling, which is available at a 40 μm isotropic voxel resolution, was upsampled to the original 20 μm resolution, using nearest neighbour interpolation. Images labelled for the different hippocampal regions were analysed to reconstruct the meshes that were adopted as anatomical landmarks for the automatic orientation of the axonal and dendritic probability function. The algorithm implemented morphological operation using the built-in function bwconncomp, regionprop2 and bwperim to identify, depending on the resulting binary mask areas, single cells or closely packed somas and connect them. Once the labelled regions were identified, the algorithm performed the 3D reconstruction of the images obtained from different planes.

### Image segmentation

The transformation of the intensity greyscale images consisted of applying rectangular region of interests of variable size corresponding to the localization of the hippocampal formation. A contrast enhancement filter responsible for remapping the image intensity values to the full display range was then performed. This procedure sharpened the differences between stained cells and background. A dynamic Otsu thresholding[56] to generate binary masks for each coronal image was then adopted. The optimal threshold was automatically selected by referring to the average intensity value of the SLM layer, which is known to be poorly populated. The estimated size of PCs soma is between 20 μm and 25 μm in diameter, each non-zero pixel was therefore associated to cell soma coordinates.

Furthermore, given the pixel size, neuronal classes could not be differentiated according to morphological features such as soma diameter. The reconstruction of the 3D neuronal placement was achieved by assigning the $x$ and $y$ coordinates as the pixel indices (rows, columns) multiplied by pixel resolution, whereas the $z$ coordinate was obtained by multiplying the index of the stacked image by vertical resolution.

The full-scale model was obtained by randomly pruning the 3D coordinates that resulted from image analyses to the putative number of PCs estimated in ref. [32] and respecting the ratio of 10% between inhibitory (Inh) and excitatory (Exc). The final neuronal population distribution was 4.8 million PCs and 480,000 inhibitory neurons (Supplementary Table 3 and Fig. 2). The overall neuronal population (Exc/Inh) was further divided in eight classes (1 Exc, 7 Inh) (Fig. 2 and Supplementary Table 1). The 3D surface of the hippocampal volume resulting from image labelling has been divided into a superficial and a deep side corresponding respectively to SR and to the SO (Fig. 2b,c) through a custom-made nearest neighbour algorithm. Subsequently, inhibitory population was divided in two main classes depending on the relative position from SO and SR calculated as the minimum Euclidean distance. The interneurons predominantly laying in SR and SLM (IVY-like and NGF-like) were shifted radially to account for the fact that image labelling has been performed by considering only SO and SP. The complete scaffolding of the neurons in the 3D volume resulted from the placement procedure was visualized with the visualization software Mayavi (v4.8.1; https://hal.science/hal-00502548).

### Neuronal density analysis

The cell density associated to the 3D neuronal distribution was obtained by adopting an Octree family algorithm that recursively partitioned the 3D points of the neuronal placement into subvolumes (300 × 300 × 300 μm³) returning the number of neurons in each voxel. The density gradient in the transversal direction was obtained by slicing the CA1 volume along medio-lateral dorso-ventral planes and analysing the cell densities in three different concentric lines running in parallel to the CA1 internal and external surfaces and approximately corresponding to SO, SP and SR. The procedure was repeated by creating

ten slices at different antero-posterior positions to create an average trend (Fig. 3c). The density gradient in the antero-posterior direction was obtained by sampling 25 voxels (white circles in Fig. 3d) from the anterior to the posterior part. Each voxel sampling was repeated 13 times to account for local density variability. This number resulted from the need to map the cell density in the whole CA1 volume which is estimated in 547 mm$^3$ (see main text). Given the size of a single voxel, this volume can be obtained with about 20 repetitions of 25 samplings. However, the morphology of CA1 is not uniform and the CA1 is progressively thinner going from the anterior to the posterior part. To account for this variation, we have reduced the number of repetitions down to 13 allowing one to cover the CA1 posterior tip with little overlap of voxels.

### Neuronal morphology

The rule to generate neuronal connection pairs was implemented assuming that neuronal classes are characterized by specific morphological properties. These properties, derived from literature[36,39] or public databases, have been modelled as geometrical probability volumes mimicking the cross-section volume of axons and dendrites. Every neuron belonging to a specific class has been associated with a series of parameters accounting for its position and its orientation with respect to the three canonical axes (transverse, longitudinal and horizontal) allowing a proper orientation. The algorithms were written in MATLAB and were identified as 'positional-morpho-anatomical' modelling[25]. All cells were associated with their relative distances from CA2, subiculum and internal subregions (landmarks identification), by calculating the minimum Euclidean distances. The CA2 and subiculum landmarks allowed the modelling of the orientations of PC and interneurons observed in the rodent CA1 hippocampus. In particular, the minimum distance vectors between CA1 neurons and CA2 surface, or between CA1 neurons and subiculum surface, were primarily used to generate the transversal plane adopted to orient the PC axonal branch extent (see 'Tubes' section).

The axonal and dendritic arborizations of PCs have been modelled as a combination of tubes (see 'Tubes' section) and cones (see 'Cones' section), respectively, while the interneuron axonal and dendritic arborizations were modelled as a combination of ellipsoids (see 'Ellipsoid' section) and cones. The variable size of the different geometrical volumes is reported in Supplementary Table 1 and rescaled by a factor of 1.5 according to the values adopted in ref. [25].

### PCs

According to experimental findings in rodents[39], the axons of PCs project into the SO to bifurcate transversally towards the subiculum with poor divergence to the CA2. We have therefore created a single tubular volume (see 'Tubes' section) to describe axonal density function of PCs since their somas lay in the SP and axons emerge from somas and project to the SO. The apical and basal dendritic probability density functions were automatically oriented in the directions projecting to SLM and SO, respectively.

### Tubes

PC axonal branches have been modelled as tubular probability volumes according to the points resulting from the intersection between the transversal planes and the deeper region of the CA1 (SO group of CA1 surface) (Fig. 4b,c). A dedicated routine has been generated to wrap a cylinder (tube) of radius $r$ along any 3D curve defined by a [3, $N$] vector of points coordinates, where $N$ varied according to the numerosity of the intersecting points. This procedure allowed the generation of a tubular axonal branch extension with custom cross-section (diameter of 300 μm) running with good approximation in parallel to the surface bending. Extending observations from animal models, PC axons projected unidirectionally towards the subiculum with a limited back-propagation to the CA2 proportional to the geodesic distance from the CA2 and with a maximum length of 500 μm.

### Cones

Apical and basal dendritic arborization of PCs have been modelled as conical point volumes with extent and orientation based on morpho-anatomical constraints (Fig. 4d). To parameterize conical probability density functions, we assumed that **u** and **v** are two orthogonal vectors that lie in the plane of the circle forming the basis of the cone. To build a cone between point $O$ (apex) and base centre point (P) with a given radius $R$, we determined the norm of the cone base plane, which is given by $d = P - O$. The probability density functions associated with the cone were then modelled as scattered tridimensional points following equation (5).

$$\begin{bmatrix} x \\ y \\ z \end{bmatrix} = \begin{pmatrix} O_x + \frac{h}{H}dx \\ O_y + \frac{h}{H}dy \\ O_z + \frac{h}{H}dz \end{pmatrix} + \begin{pmatrix} R \cdot \frac{h}{H} \cdot \cos \vartheta \cdot u_x \\ R \cdot \frac{h}{H} \cdot \cos \vartheta \cdot u_y \\ R \cdot \frac{h}{H} \cdot \cos \vartheta \cdot u_z \end{pmatrix} + \begin{pmatrix} R \cdot \frac{h}{H} \cdot \sin \vartheta \cdot v_x \\ R \cdot \frac{h}{H} \cdot \sin \vartheta \cdot v_y \\ R \cdot \frac{h}{H} \cdot \sin \vartheta \cdot v_z \end{pmatrix} \quad (1)$$

$$0 \le h \le H, 0 \le \vartheta \le 2\pi$$

where $H = |P - O| = d$.

Basal and apical dendrites were oriented towards the directions connecting cell placement with minimum distance to deep (SO) and superficial (SR) CA1 surface, respectively.

### Interneurons

According to the variability of rodent interneuron morphologies[36,37], we have assumed seven classes of GABAergic cells representing clusters of different interneurons sharing analogous morphological features. The choice of seven classes was dictated by data availability in public repositories and published articles that were used to calculate morphological parameters (Supplementary Table 1). In particular, the modelled classes are representative of 11 different interneuron subtypes described below:

**Perisomatic-like.** These cells, which can be traced back to rodents PV$^+$ and CCK$^+$ basket cells and axo-axonic cells[36], have somas laying in the SP, axonal cloud projecting within the SP and dendrites crossing the entire CA1 from SO to SLM. The axon was modelled as an ellipsoid, while both apical and basal dendrites were modelled as two cones (Supplementary Fig. 1 and Supplementary Table 1). Cell somas were distributed randomly in the CA1 volume.

**OLM-like.** These cells, which can be traced back to rodents SO-OLM and back-projecting cells, have somas confined in the SO. The axon projects a thin filament to the SLM where generates a dense plexus while the dendrite spreads in the SO[36]. A combination of ellipsoids was used to model both axons and dendrites (Supplementary Fig. 1 and Supplementary Table 1). Cell somas were selected in the outer part of the SO interneuron subgroup.

**IVY-like.** These cells, which can be traced back to rodents IVY and bistratified cells, have cell bodies mainly distributed within SP and SR, but they also populate SO[36]. Axons and dendrites are predominantly located within SR, SP and SO with protrusion within the SLM for cells located in the superficial SR[36]. Furthermore, dendrites are preferentially confined inside axonal clouds. Single ellipsoids have been used to model both neurites (Supplementary Fig. 1 and Supplementary Table 1), while cell somas were selected in the outer part of the SR interneuron subgroup.

**TRI-like.** These cells, which can be traced back to rodents trilaminar cells, have somas in the SO with axons crossing CA1 layers[36] and dendrites preferentially confined in the proximity of the soma[36]. Ellipsoids were used to model both axons and dendrites (Supplementary Fig. 1 and Supplementary Table 1). Cell somas were selected in the SO interneuron subgroup.

**SCA-like.** These cells, which can be traced back to rodent Schaffer collateral-associated cells, have somas in the SR, axons projecting to the SO and dendrites projecting to the SLM[36]. Eccentric ellipsoids have been used to model both axons and dendrites. Cell somas were positioned by selecting 3D coordinates from the outer part of the SR subgroup, subsequently somas were shifted towards the SLM direction proportionally to the distance between soma and SR border (Supplementary Fig. 1 and Supplementary Table 1).

**PPA-like.** These cells, which can be traced back to rodent perforant pathway-associated cells, have somas in the SR, axons are confined in the SR and SLM and dendrites project in both directions and extend towards the SP[36,57]. A large ellipsoid was used to model the axon, and two cones were adopted for dendrites (Supplementary Fig. 1 and Supplementary Table 1). Cell somas were selected from the outer part of the SR interneuron subgroup and shifted towards the SLM direction proportionally to the distance between soma and SR border.

**NGF-like.** These cells, which can be traced back to rodent neurogliaform, represented the most abundant GABAergic population in the whole CA1. They have somas and a short dendritic tree well confined in the SLM. Conversely, a large axon is directed towards the SO with projections in the SR[39,58]. Ellipsoids were used to represent both axonal and dendritic probability density functions (Supplementary Fig. 1 and Supplementary Table 1). Cell somas were selected in the outer part of the SR interneuron subgroup and shifted towards the SLM direction proportionally to the distance between soma and SR border. The shifting length was larger than the one adopted for neurons in the SR.

The sizes of all the probability density functions (Supplementary Fig. 1) have been generated by calculating the average rodent axonal and dendritic extension from literature[36,37] and from public repositories (www.neuromorpho.org). A 1.5 scale factor was introduced to compensate for the differences observed between rodents and human PC dendrites. A normal distribution for each parameter describing cones and ellipsoids was then generated and parameters were randomly sampled from the distribution to account for cell-to-cell variability.

### Ellipsoid

Assuming that any quadratic function $f(x_1,…,x_n)$ can be written in the form $X^TQX$, where $Q$ is a symmetric matrix ($Q = Q^T$), given a system of eigenvectors (unit vectors) that diagonalize the symmetric matrix, any ellipsoid can be described as a volume oriented in the direction set by the eigenvectors and elongated along the semi-axis as set by the eigenvalues.

The probability ellipsoid representing axonal projections can thus be easily parametrized considering an orthonormal system of eigenvectors $\mathbf{v_1}, \mathbf{v_2}, \mathbf{v_3}$ associated, respectively, with the eigenvalues $\lambda_1, \lambda_2, \lambda_3$ of a $3 \times 3$ symmetric positive matrix M. If

$$V = [v1, v2, v3] \qquad (2)$$

Then

$$V^TMV = \begin{bmatrix} \lambda_1 & 0 & 0 \\ 0 & \lambda_2 & 0 \\ 0 & 0 & \lambda_3 \end{bmatrix} = D[\lambda_1, \lambda_2, \lambda_3]; \qquad (3)$$

$V^TMV$ is a diagonal matrix containing the eigenvalues of M and the normalized vectors $\mathbf{v_1}, \mathbf{v_2}, \mathbf{v_3}$ are called the principal axis of M.

Given an arbitrary base of orthonormal vectors $\mathbf{u_1}, \mathbf{u_2}, \mathbf{u_3}$ (the orientation vectors) determining the matrix U and a diagonal matrix (D) of arbitrary eigenvalues (semi-axis lengths), it is possible to obtain the symmetric matrix Q with equation (3).

$$Q = UDU^T \qquad (4)$$

The orientation vectors were created by calculating the relative positioning of CA1 neurons with respect to other hippocampal regions. Transversal planes were generated as planes containing the minimum distance vectors connecting CA1 somas with CA2 and subiculum mesh points (Fig. 4).

The ellipsoidal probability density function was modelled as scattered points according to the canonical parametric equations:

$$x = \lambda_1 \cos\vartheta \sin\varphi, y = \lambda_2 \cos\vartheta \sin\varphi, z = \lambda_3 \cos\varphi \qquad (5)$$

where $0 \leq \vartheta < 2\pi$ and $-\pi \leq \varphi \leq 0$.

The points composing the ellipsoid probability density function were obtained by generalizing the canonical parametric equations to an arbitrary orientation according to the calculated vectors.

Axons were converted into convex hulls, whereas dendrites into a variable number of scattered points. Each point represented a volume of about 64,000 $\mu m^3$, corresponding to a 40 $\mu m$ side voxel. For example, the cones adopted to model apical dendrite of PCs (Supplementary Table 1) had an average volume of about 11,780,000 $\mu m^3$, yielding a total of 184 points (11,780,000/64,000).

### Neuronal connectivity

Neuronal connectivity was performed between neurons belonging to a pre- and a post-synaptic class. The axonal and dendritic probability density functions were preliminarily circumscribed within their minimal bounding boxes and to reduce the computational effort, the pre-synaptic neuron was intersected only against neurons whose dendritic bounding boxes overlapped with its axonal bounding box (Extended Data Fig. 1). The overlapping between axonal and dendritic bounding boxes ($BB1_{3D}, BB2_{3D}$) was preliminarily determined by applying the following set of equations:

$$BB1_{3D} = (x : (x_{min1}, x_{max1}), y : (y_{min1}, y_{max1}), z : (z_{min1}, z_{max1})) \qquad (6)$$

$$BB2_{3D} = (x : (x_{min2}, x_{max2}), y : (y_{min2}, y_{max2}), z : (z_{min2}, z_{max2})) \qquad (7)$$

$$\text{overlap3D}(BB1_{3D}, BB2_{3D}) =$$

$$\text{overlap1D}(BB1_{3D}.x, BB2_{3D}.x) \&$$

$$\text{overlap1D}(BB1_{3D}.y, BB2_{3D}.y) \& \qquad (8)$$

$$\text{overlap1D}(BB1_{3D}.z, BB2_{3D}.z)$$

where, given $BB1_{1D} = (x_{min1}, x_{max1})$ and $BB2_{1D} = (x_{min2}, x_{max2})$

$$\text{overlap1D}(BB1_{1D}, BB2_{1D}) = x_{max1} \geq x_{min2} \& x_{max2} \geq x_{min1}$$

The intersection was evaluated only on neurons with overlapping bounding boxes. Connection pairs were calculated through an iterative algorithm assessing the inclusion of at least one dendritic point into the axonal convex hull. The final number of connection pairs was obtained following a pruning procedure which was performed according to the number of estimated contacts between the two neuronal classes. This number was adapted from rodents and was obtained by multiplying the synaptic connection probability and the total number of neurons composing the two classes (see hippocampome.org, Supplementary Table 2).

The total time required to estimate the connectivity matrix depended on (1) the number of points in each dendritic probability

density function, (2) the number of potential intersections and (3) the numerosity of each neuronal class. Only the first parameter was set a priori and could be adjusted to limit the computation time.

The positional-morpho-anatomical modelling algorithm was parallelized to run on a supercomputer, and the full network was generated in a period of 240 h on 20 CPUs. The CA1 network with 5.28 million neurons generated ~40 billion synapses, and it was created on the Lyra server available at labcsai (http://www.labcsai.unimore.it) and equipped with an Intel Xeon 20 core 6230 2.1 GHz with 40 processors.

### Data analysis
The similarity between distributions has been estimated with the KL divergence method, quantifying how much one probability distribution differs from another probability distribution. Given the distributions P and Q, the KL divergence can be calculated as the negative sum of probability of each event in P multiplied by the log of the probability of the event in Q over the probability of the event in P. The KL divergence score is large when the probability for an event from P is large, but the probability for the same event in Q is small, there is a large divergence.

Statistics are reported as mean ± standard error of the mean (s.e.m.) unless otherwise specified.

### Reporting summary
Further information on research design is available in the Nature Portfolio Reporting Summary linked to this article.

## Data availability
Source data for Figs. 3–5 are available with this manuscript. Source images are available from the BigBrain repository[29]. The scaffold model resulting from the analysis of the images is available in the EBRAINS knowledge graph[59] in the form of a text file with Global Identification (GID) numbers and $x,y,z$ coordinates that can be used to reproduce Figs. 2 and 6. At the same link[59] the connection pairs can be downloaded as a collection of multiple text files containing the GID numbers of the presynaptic and postsynaptic neurons and a hdf5 file was generated to allocate each txt file as an hdf5 key.

## Code availability
The codes allowing to implement the computational pipeline for a subset of PCs and interneurons are released as a Code Ocean capsule[60].

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

## Acknowledgements

This work has received funding from the EU Horizon 2020 Framework Program for Research and Innovation (Specific Grant Agreement 945539, Human Brain Project SGA3) to M.M. and E.D., the Flag ERA JTC 2019 (MILEDI project to M.M. and SMART-BRAIN project to J.M.), the French National Research Agency (ANR) as part of the second 'Investissements d'Avenir' program, ANR-17-RHUS-0004, EPINOV (https://anr.fr) to V.J., and from the National Recovery and Resilience Plan (NRRP), Mission 4, 'Education and Research' Component 2, 'From research to Business' Investiment 3.1 – Call for tender No. 3264 of Dec 28, 2021 of Italian Ministry of University and Research funded by the European Union – NextGenerationEU – Award Number: Project code IR0000011, Concession Decree No. 117 of June 21, 2022 adopted by the Italian Ministry of University and Research, CUP B51E22000150006, Project title 'EBRAINS-Italy (European Brain ReseArch INfrastructureS-Italy)' to J.M., M.M. and E.D. FENIX computing and storage resources were granted under the Specific Grant Agreement No. 800858 (Human Brain Project ICEI), from the Swiss National Supercomputing Centre (CSCS) under project ID ich011, and CINECA (Italy) to M.M. Editorial support was provided by Annemieke Michels of the Human Brain Project. The authors thank L. Kusch (Aix Marseille University) for the technical assistance in compiling the NEST simulator in a docker image.

## Author contributions

Conceptualization, D.G., J.M. and M.M.; methodology, D.G., J.M., P.T. and S.M.G.S.; writing—original draft, D.G., J.M. and M.M.; writing—review and editing, D.G., J.M., P.T., S.M.G.S., E.D., V.J. and M.M.; funding acquisition, J.M., M.M., V.J. and E.D.; resources, J.M., M.M. and E.D.; supervision, M.M. and E.D.

## Competing interests

The authors declare no competing interests.

## Additional information

**Extended data** is available for this paper at https://doi.org/10.1038/s43588-023-00417-2.

**Correspondence and requests for materials** should be addressed to Daniela Gandolfi, Jonathan Mapelli or Michele Migliore.

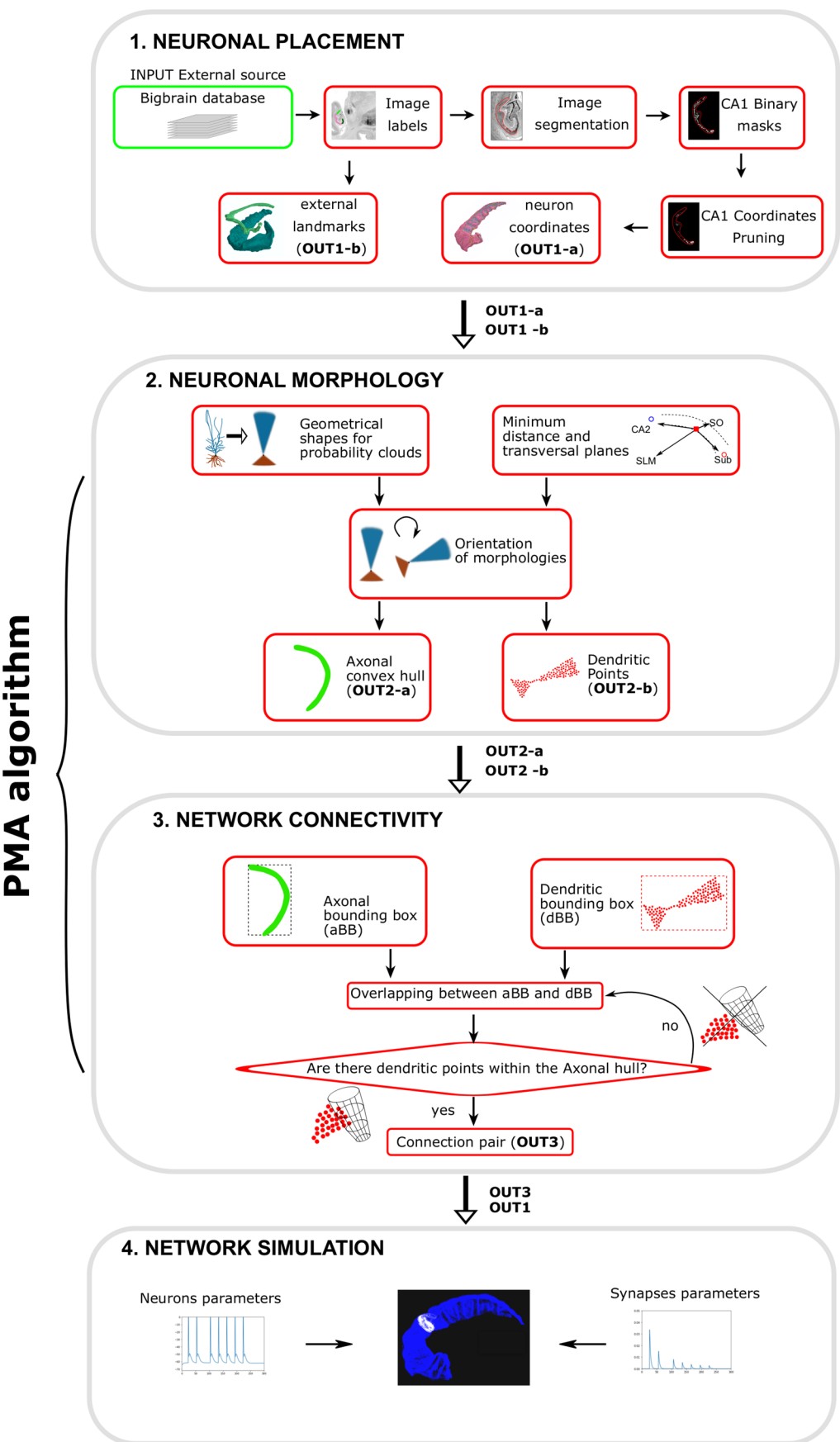

Extended Data Fig. 1 | See next page for caption.

**Extended Data Fig. 1 | Workflow for the generation of human CA1 circuit.**
Gray boxes represent the adopted procedural blocks. Red boxes represent computational functions. Green box represents non elaborated data obtained from public repositories (external source). 1 Neuronal placement: Cells coordinates have been downloaded from the BigBrain image database in the form of grayscale images. Images have been segmented and hippocampus subregions have been isolated (CA1, CA2, CA3, CA4, Subiculum). Putative cell soma positions have been isolated through image analysis. Subsequently, a pruning procedure allowed to assign the 3D coordinates to excitatory (1) and inhibitory (7) classes according to their relative positions in CA1 layers (OUT1a). CA2 and Subiculum surface meshes have been generated (OUT1b). 2 Neuronal morphologies: Geometrical shapes were generated by assigning morphological parameters to dendritic and axonal features. Neurons were oriented according to external landmarks and axonal convex hulls (OUT2a) and dendritic point volumes (OUT2b) were generated. 3 Network connectivity; Bounding boxes for axons and dendrites were generated and an iterative algorithm calculated the intersections of axonal hull with dendrites showing overlapping bounding boxes. A connection pair (OUT3) was generated when at least one dendritic point fell into the axonal hull. 4 Network simulation. The scaffold and connectivity matrix were loaded into NEST simulator to simulate CA1 activity. The PMA algorithm is represented by blocks 2 and 3.

# nature research

# Reporting Summary

Nature Research wishes to improve the reproducibility of the work that we publish. This form provides structure for consistency and transparency in reporting. For further information on Nature Research policies, see our Editorial Policies and the Editorial Policy Checklist.

## Statistics

For all statistical analyses, confirm that the following items are present in the figure legend, table legend, main text, or Methods section.

| n/a | Confirmed | |
|---|---|---|
| ☐ | ☒ | The exact sample size (*n*) for each experimental group/condition, given as a discrete number and unit of measurement |
| ☐ | ☒ | A statement on whether measurements were taken from distinct samples or whether the same sample was measured repeatedly |
| ☒ | ☐ | The statistical test(s) used AND whether they are one- or two-sided<br>*Only common tests should be described solely by name; describe more complex techniques in the Methods section.* |
| ☒ | ☐ | A description of all covariates tested |
| ☒ | ☐ | A description of any assumptions or corrections, such as tests of normality and adjustment for multiple comparisons |
| ☐ | ☒ | A full description of the statistical parameters including central tendency (e.g. means) or other basic estimates (e.g. regression coefficient) AND variation (e.g. standard deviation) or associated estimates of uncertainty (e.g. confidence intervals) |
| ☒ | ☐ | For null hypothesis testing, the test statistic (e.g. *F*, *t*, *r*) with confidence intervals, effect sizes, degrees of freedom and *P* value noted<br>*Give P values as exact values whenever suitable.* |
| ☒ | ☐ | For Bayesian analysis, information on the choice of priors and Markov chain Monte Carlo settings |
| ☒ | ☐ | For hierarchical and complex designs, identification of the appropriate level for tests and full reporting of outcomes |
| ☒ | ☐ | Estimates of effect sizes (e.g. Cohen's *d*, Pearson's *r*), indicating how they were calculated |

*Our web collection on statistics for biologists contains articles on many of the points above.*

## Software and code

Policy information about availability of computer code

| Data collection | No software used to collect data |
|---|---|
| Data analysis | The codes allowing to implement the computational pipeline for a subset of Pyramidal cells and interneurons will be released as a Code Ocean capsule (https://doi.org/10.24433/CO.8325351.v1). Codes employed in this research have been developed though customized procedures mainly in the MATLAB (v2019b) environment.<br>The computational pipeline relies also on the following software/packages:<br>-Python 3.6.13<br>-Myavi 4.8.1<br>-NEST 3.0<br>-Excel (Windows 10)<br>-Jupyter Notebook 6.5.2<br>-Numpy 1.24.2<br>-Vsimpl 1.5.2 |

For manuscripts utilizing custom algorithms or software that are central to the research but not yet described in published literature, software must be made available to editors and reviewers. We strongly encourage code deposition in a community repository (e.g. GitHub). See the Nature Research guidelines for submitting code & software for further information.

## Data

Policy information about availability of data

All manuscripts must include a data availability statement. This statement should provide the following information, where applicable:

- Accession codes, unique identifiers, or web links for publicly available datasets
- A list of figures that have associated raw data
- A description of any restrictions on data availability

Source data for Figure 3-5 are available with this manuscript. Source images are available from the BigBrain repository. The scaffold model resulting from the analysis of the images is available in the EBRAINS knowledge graph in the form of a text file with GIDs number and x,y,z coordinates that can be used to reproduce Figures 2 and 6. At the same link the connection pairs can be downloaded as a collection of multiple text files containing the GIDs of the presynaptic and postsynaptic neurons and a hdf5 file was generated to allocate each txt file as an hdf5 key.

# Field-specific reporting

Please select the one below that is the best fit for your research. If you are not sure, read the appropriate sections before making your selection.

☒ Life sciences    ☐ Behavioural & social sciences    ☐ Ecological, evolutionary & environmental sciences

For a reference copy of the document with all sections, see nature.com/documents/nr-reporting-summary-flat.pdf

# Life sciences study design

All studies must disclose on these points even when the disclosure is negative.

| | |
|---|---|
| Sample size | No sample size was chosen since the full image database was considered to perform image analysis |
| Data exclusions | No data were excluded from our research |
| Replication | No replication has been performed since only a single image dataset was available |
| Randomization | Our study does not compare a positive and a control group, randomization is therefire not relevant to this study. A randomization procedure has been performed to connect neurons in an alternative fashion in respect to our approach. |
| Blinding | Our study does not include trials based on the concealment of group allocation of individuals |

# Reporting for specific materials, systems and methods

We require information from authors about some types of materials, experimental systems and methods used in many studies. Here, indicate whether each material, system or method listed is relevant to your study. If you are not sure if a list item applies to your research, read the appropriate section before selecting a response.

## Materials & experimental systems

| n/a | Involved in the study |
|---|---|
| ☒ ☐ | Antibodies |
| ☒ ☐ | Eukaryotic cell lines |
| ☒ ☐ | Palaeontology and archaeology |
| ☒ ☐ | Animals and other organisms |
| ☒ ☐ | Human research participants |
| ☒ ☐ | Clinical data |
| ☒ ☐ | Dual use research of concern |

## Methods

| n/a | Involved in the study |
|---|---|
| ☒ ☐ | ChIP-seq |
| ☒ ☐ | Flow cytometry |
| ☒ ☐ | MRI-based neuroimaging |

