## [Peer Review File · Nature Computational Science]

Peer Review Information

Journal: Nature Computational Science

Manuscript Title: Full-scale scaffold model of the human hippocampus CA1 area

Corresponding author name(s): Daniela Gandolfi, Jonathan Mapelli, Michele Migliore

Editorial Notes:

Redactions – published data Parts of this Peer Review File have been redacted as indicated to remove third-party material.

Reviewer Comments & Decisions:

Decision Letter, initial version:
--

Dear Dr Gandolfi,

Thank you for submitting your manuscript "Full-scale scaffold model of the human hippocampus CA1 area". I am pleased to tell you that we are sending your paper out for formal peer review. Before we can do so, please read the below carefully as we require a few documents.

If you have not done so already, please alert us to any related manuscripts from your group that are under consideration or in press at other journals, or are being written up for submission to other journals (see <https://www.nature.com/authors/policies/duplicate.html> for details).

We are asking all corresponding authors of primary research articles to complete an Editorial Policy Checklist that verifies compliance with all required editorial policies. The form should be completed and returned within 48 hours, if you have not already done so. Please note that the form is a dynamic 'smart pdf' and must therefore be downloaded and completed in Adobe Reader. We will then flatten them for ease of use by the reviewers. If you would like to reference the guidance text as you complete the template, please access these flattened versions at <https://www.nature.com/authors/policies/availability.html>

Editorial Policy Checklist: <https://www.nature.com/documents/nr-editorial-policy-checklist.zip>

We also ask that you complete a reporting summary that collects information on experimental design: Reporting summary: <https://www.nature.com/documents/nr-reporting-summary.zip>

In addition, as your paper relies on code that is central to the main claims, we will ask the reviewers to evaluate the code during the peer review process (for more details on this please see this editorial in Nature <https://www.nature.com/articles/d41586-018-02741-4>).

Reproducibility and re-usability of code are very important to us so, to facilitate this process, we are currently running a trial in partnership with Code Ocean to enable authors to share fully-functional and executable code accompanying their articles and to facilitate peer review of code by the reviewers (for more details please see <http://blogs.nature.com/ofschemesandmemes/2018/08/01/nature-research-journals-trial-new-tools-to-enhance-code-peer-review-and-publication>). We expect this functionality to speed up the peer review of your paper as it will facilitate the reviewer's assessment.

The use of the Code Ocean platform for peer review of code associated with this paper will be under the same confidentiality and anonymity agreements as the rest of manuscript materials.

Code Ocean is a cloud-based reproducibility platform where authors upload code and data and configure the necessary computational environment for reproduction. The code, data, metadata, and computational environment -- called a 'compute capsule' -- can then be accessed by reviewers in an anonymous fashion, and upon publication, provided to readers via a link from the article. Code Ocean supports all open source programming languages, as well as Stata and MATLAB and compute capsules can be created from existing GitHub folders by easy drag and drop.

Code Ocean staff will assist you in generating a compute capsule for your code and a working copy of this compute capsule will be used in the peer review process (after a brief review by Code Ocean staff, to ensure that everything runs). If you have selected Double Blind Peer Review, we will make sure the capsule contains no information about your identity.

If your code is accepted for publication, Code Ocean will assign a Digital Object Identifier (DOI) to your compute capsule. It will then be embedded into your article. Code Ocean, through CLOCKSS, will guarantee the preservation of all elements of Code Ocean's compute capsules, including the code, data, results, metadata, Dockerfile and Docker image (computational environment) associated with your paper.

By using this platform, other researchers will then be able to easily find and run the code, as well as build upon your work, without any additional setup or configuration of the software. It will also enable preservation of your code, data and the complete environment so that the code associated with this publication is maintained. Should the paper be rejected, you will retain full control over the compute capsule, and be able to decide what to do with it (publish it, modify it etc).

We very much hope you will be interested in engaging in this trial. Please let us know as soon as possible if you wish to participate and we will provide you with further guidelines for setting up the compute capsule. An overview of the process can be found here: <https://help.codeocean.com/publishing-on-code-ocean/peer-review-on-code-ocean> .

Alternatively, if you do not want to engage in this trial, or if Code Ocean is not a good fit (<https://help.codeocean.com/en/articles/3294415-what-is-and-is-not-a-good-fit-for-publishing-and-sharing-on-code-ocean>), we ask you to complete the following Software and custom code submission checklist:

Software supplement: <https://www.nature.com/documents/nr-software-policy.pdf>
(Please note that the form is a dynamic 'smart pdf' and must therefore be downloaded and completed in Adobe Reader.)

To improve transparency in authorship we are requesting that all authors identified as 'corresponding author' on published papers create and link their Open Researcher and Contributor Identifier (ORCID) with their account on the Manuscript Tracking System (MTS), prior to acceptance. ORCID helps the scientific community achieve unambiguous attribution of all scholarly contributions. You can create and link your ORCID from the home page of the MTS by clicking on 'Modify my Springer Nature account'. For more information please visit www.springernature.com/orcid.

If you wish to send us revised manuscript files as a result of filling out the checklists, you may email them to us at the same time.

Please note that *Nature Computational Science* is a Transformative Journal (TJ). Authors may publish their research with us through the traditional subscription access route or make their paper immediately open access through payment of an article-processing charge (APC). Further information regarding *Nature Computational Science* publishing options and our APC is available [here](https://www.springernature.com/gp/open-research/policies/journal-policies).

For submissions from January 2021, if your paper is accepted for publication in *Nature Computational Science*, you will be asked to choose the publishing option that works best for you. If your research is supported by a funder that requires immediate open access (e.g. according to Plan S principles) then you should select the gold OA route, and we will direct you to the compliant route where possible. For authors selecting the subscription publication route our standard licensing terms will need to be accepted including our self-archiving policies. Those standard licensing terms will supersede any other terms that the author or any third party may assert apply to any version of the manuscript.

Finally, we encourage you to share a preprint of the original submitted version of your paper so as to minimize delays in communicating your research findings; benefits of preprints include early visibility, and citations (<https://www.natureindex.com/news-blog/preprints-boost-article-citations-and-mentions>) and demonstration of research progress. You may want to consider the multidisciplinary Research Square preprint platform (<https://www.researchsquare.com/browse>), provided by our partner Research Square, where your preprint will be publicly available with a citable DOI under a CC-BY license. You are of course free to use a discipline-specific preprint platform of your choice. More information about our preprint policy can be found in the following link: <https://www.nature.com/nature-research/editorial-policies/preprints-and-conference-proceedings#preprints>

Please note that *Nature Computational Science* implements transparent peer review of original research manuscripts, in which we publish the reviewer comments to the authors, author rebuttal letters and editorial decision letters as a supplementary peer review file, if the author agrees at the point of acceptance. This will apply to new manuscripts submitted on or after 17th Feb 2021. Upon author request, confidential information and data can be removed from the reviewer reports and

rebuttal letters prior to publication. For more information, please refer to our [FAQ page](https://www.nature.com/documents/nr-transparent-peer-review.pdf).

Thank you very much for your attention to this. We look forward to hearing from you about the Code Ocean trial and receiving the policy checklist and reporting summary, but please let me know if you have any questions.

Best regards,

Ananya Rastogi, PhD
Associate Editor
Nature Computational Science

Author Rebuttal to Initial comments

We thank all reviewers for the useful and insightful comments to improve the manuscript, which has been substantially revised to consider all the points raised, as described below.

Reviewer #1

-I think the general procedure/workflow for generating this scaffold model is at least as important as the resulting model itself which depends on several assumptions. At present I find it hard to get an overview over this workflow.

- I suggest the authors add a figure/diagram illustrating the workflow, and give an overview over what assumptions are made at each step

Good point. We have now included more details on the implementation. The new Figure 1, including a schematic representation of the workflow, should now help the reader to get an immediate overview of the pipeline used to build the scaffold model, from image analysis to the network simulation. The main text has been revised to outline the assumption used at each step.

- Some steps involve choices of parameters (e.g., cone widths) which should be specified in a table so that readers get a clear overview over which parts of the procedure depends only on imaging data and which on choices of the modeller.

A table containing the parameter adopted to model the cell morphologies is reported in Table SM1. In the main text we now also outline the rationale and the assumptions used to choose parameters for morphological and connectivity properties and we have clarified in the text that table SM-1 pertains to neuronal morphology and network connectivity while parameters adopted for the image analysis are reported in separate sections of Results and Methods

- The paper presents a single model where Figure 6 shows validation results. This is a bit unsatisfactory as it is unclear how sensitive the resulting model is to parameter choices. It would be great if the authors could expand on how the resulting models from the workflow depend on the various choices having been made.

To consider this point, in Figure SM-4 we report the results of changing pyramidal morphological parameters. Connectivity was evaluated from 9 instances of a network composed of 200,000 PCs randomly selected from the full model. Kullback-Leibler divergence method estimated that the shape of the probability densities was not significantly affected in all the tested configuration (average KL-score 0.012 range 0.0016-0.042 compared to 0.7, obtained from the comparison with randomly connected network).

- The main reason for constructing scaffold models is to use them as a starting point for making dynamical networks. Here this application is exemplified by a simulation using integrate-and-fire neurons. However, no results are shown. While I understand that this simulation was for demonstration purposes only, I think it would be nice if some simulation results were shown in the paper.

Following this suggestion, we have added demo simulations obtained with the purely excitatory network and full network. Accordingly, a new figure 7, Figs. SM5-6 and two supplementary movies have been added. In the main text we stress that these are demo simulations, with the exclusive purpose to qualitatively test the network integrity, and to provide an estimate of the simulation costs required to simulate a network of this scale.

- Around Figure 1 the authors discuss co-simulation and neural mass theory, but it is unclear to

me how this is related to the results of the paper. Neural masses typically refer to populations of neurons, while the "scaffold model" has single neurons as the basic unit.

We agree with the reviewer that introducing this aspect in the first figure of the work can be somewhat misleading. We have now moved the original Fi. 1 to the supplemental Material, as Fig. SM-1. We feel that mentioning the co-simulation with a neural mass whole-brain model such as TVB (which will be described in a future work), is important to introduce the framework in which our full-scale models of brain regions will be eventually plugged-in to greatly increase the whole-brain model resolution.

Reviewer #2

We thank the reviewer for the thorough examination of the manuscript. Point by point responses can be found below.

1. I appreciate that data sparseness forces modelers to make many assumptions and generalizations (e.g., from rodent data to human). This is totally acceptable as far as assumptions and generalizations are clearly exposed to the reader. First of all, this allows scientific reproducibility but also revisions and improvements of the model. In the manuscript, several parameters and assumptions are given without precise references. I can provide few examples, but I encourage you to check the other parameters used by the model.

1.1. In lines 260, 261, 600, you said that the axon probability cloud is modelled with a tubular structure of radius 300 μm and it can project backwards of minimum 150 μm and maximum of 500 μm , but the numbers are not justified.

More information on the assumptions for the morphological parameters and connectivity used in building the model have been added to the relevant paragraphs in the revised version of the manuscript (see Methods). Briefly, in lines 260-61-600 (of the first submitted version of the manuscript) we didn't refer to a specific literature since, to our knowledge, published experimental data on human pyramidal axons are still missing. We fully agree with the reviewer that is important to underline that human morphology parameters have been derived from rodent literature

(<https://neuromorpho.org/>, <http://ml-neuronbrowser.janelia.org/>) applying a scaling factor of 1.5 and taking into account the experimental evidence that pyramidal axons preferentially branch towards the Subiculum with a scarce backpropagation to the CA3. The choice of a 1.5 factor was motivated by the fact that analyzing the data reported in Benavides-Piccione et al. 2020 the average extension of human dendritic trees is approximately 1.5 the extension of the mouse dendritic trees. In the current revision process we also report the effects of adopting different scaling factors (0.75, 1.5, 3) on the network connectivity (indegree, outdegree and distances distributions).

1.2. There is no reference for all the numbers used to constrain the clouds and reported in table SM-1. In the text, you mentioned the review from Pelkey et al. 2017. I checked few interneurons, and I did not find any estimates for the clouds. In any case, even if you took parameters from rodent literature you need to take into account also a scaling factor to move to humans and this is not mentioned in the text.

In the revised version we discuss our choice for a scale factor for human neurons and interneurons morphology with respect to a rodent.

In line 685 of the first submitted version we wrote “*The probability clouds dimensions (Suppl. Fig. SM-2) of the whole dataset have been generated by taking the average size of axonal extensions and dendritic arborizations obtained from the rodent literature and rescaled of a 1.5 factor*”, however we agree with the reviewer that it is important to further clarify in the text that we adopted a motivated scaling factor to move from rodents to human. In absence of experimental data on human interneurons we refer to Pelkey et al 2017, Bezair and Soltesz 2013, Benavides_piccione et al 2020 as experimental benchmarks to set the rescaled human dendritic and axonal parameter extensions reported in Table SM1. We have analyzed data reported in literature and in public repositories and rescaled them with 1.5 factor derived from pyramidal cells (Benavide-Piccione et al 2020).

We have now inserted in the Results an explanation of the use of a scale factor.

1.3. Some of the numbers seem to be only loosely constrained by experimental data. For example, in lines 194, 512-3, you said you used a ratio between pyramidal cells and interneurons of 10% and reported the sources <https://bbp.epfl.ch/nexus/cell-atlas/> and www.hippocampome.org. Anyway, they use a ratio of 6.5% and 10.8%.

The relevant paragraph in the *Neuronal Placement* Section of the Results has been reworded to better justify our motivation to choose an I/E ratio of 10%. This is in agreement with the range of inhibitory interneurons representativeness reported for instance in Pelkey et al. 2017 and Bezaire and Soltesz 2013. Moreover, a 10% I/E ratio represents an intermediate excitatory/inhibitory ratio between the value reported in the experimental databases such as <https://bbp.epfl.ch/nexus/cell-atlas/> and more recent findings (<https://www.pnas.org/doi/full/10.1073/pnas.2018459118>) reporting an E/I ratio of approximately 20-30%.

2. My first point refers to the lack of references or justifications for model parameters. I found something similar for model assumptions. The most relevant example is the choice of the 7 subclasses of interneurons. Is there anything special about the 7 interneuron classes considered?

Without a precise justification, this choice may seem arbitrary and lead to some contradictions. For example, you chose trilaminar cells that are relatively rare in the rodent, but not interneuron-specific cells that are more abundant (respectively 700 and 7000 according to Bezaire and Soltesz 2013). You mentioned de Lanerolle et al (2013) when discussing the 8 classes of neurons (1 excitatory + 7 inhibitory) but the book chapter still does not seem to support this choice.

Considering you are building this model as a proof of concept and considering point 3 below, I am wondering if it is not a safer option to include a generic interneuron in all the four layers rather than trying to model specific interneuron types.

The 7 interneuron classes that we modelled were intentionally selected to account for the variety of interneuron morphologies described in rodent literature [e.g. Pelkey et al 2017, Bezaire and Soltesz 2013]. Each class represents a cluster of different interneuron subtypes sharing analogous morphological feature. More specifically, we modelled the morphology class of the Trilaminar cells populating the stratum oriens/alveus border, the morphology class of Perisomatic cells (PV- BASKET, Axo-Axonic and CCKBC-BASKET) populating the SP, the morphology class of Ivy, Bistratified and Interneuron specific cells populating the SR, the morphology class of Schaffer Collateral Associated cells populating the SR, the morphology class of Perforant pathway Associated cells populating the SR and the morphology class of the Neurogliaform cells populating the SLM. The seven morphological classes that we modelled are thus representative of 10 different interneuron subtypes. In our opinion, the work hypothesis of clustering interneurons depending on their morphological features was also supported by recent findings showing that human GABAergic hippocampus interneurons exhibit strong similarities with GABAergic mouse cell types and that they could be clustered into 8 classes (1 exc 7 inhib) depending on expression patterns of marker genes, and global transcriptional similarity [Davila et al 2021]. We noticed a mistake in line 416 in citing a reference that has been now emended.

3. As you mentioned in the paper, one of the striking differences between rodents and humans is the stratum pyramidale and how the pyramidal cells are distributed inside it. Anyway, you have not mentioned one particular aspect of this story, that is distribution of pyramidal cell neurites within the volume.

In rodents, the pyramidal cells have the soma and few dendrites in stratum pyramidale, while the basal dendrites are in stratum oriens, apical dendrites in stratum radiatum and apical tufts in stratum lacunosum-moleculare. In human, the pyramidal cells have most of the neurites in the big stratum pyramidale (Benavides-Piccione et al. 2019).

This challenges the simple generalization from rodent to human, and it needs to be deeply discussed. Let's consider the different classes of interneurons. In rodents, different classes of interneurons target specifically different portions of pyramidal cells contributing to specific computations. For example, PPA and OLM cells target PC apical tufts in SLM, where the pyramidal cells receive inputs from the perforant pathway. SCA cells target PC dendrites in SR where pyramidal cells receive input from Schaffer collaterals. Now, what is the meaning of modeling interneurons like PPA, OLM, and SCA as having the same neurite distribution as above if pyramidal cell dendrites do not reach SLM and SR in humans? Maybe, in humans, the above interneurons have different morphologies so that the same PC targeting is preserved. Do we have any evidence?

Naturally the position of the pyramidal cells has implications not only in the connectivity from interneuron to pyramidal cells, but also from pyramidal cells to interneurons and among pyramidal cells. It would be useful to see the properties of the connections among the 8 different neuron types. Looking at the dendritic and axonal distributions, it seems that most of the pathways may be not viable.

Now, let me come back to my second point. Do you have a clear advantage in having the 7 subclasses of interneurons compared to having a generic interneuron type?

A new paragraph in the discussion (from rodents to human) has been added in discussion and we specifically consider the 3rd point raised by the reviewer from line 551 aspect to consider the tr.

We agree with the reviewer that a modelling strategy based on the generalization from rodent to human needs to be better discussed. In absence of evidence about the real morphology of interneurons we profusely debated on the point spotted by the reviewer: it is more biased to model a generic interneuron class type or to model a few classes of interneurons with features translated from the rodent? This question opens another question: if we decide to model a single generic interneuron type, which morphological features should we adopt? Our opinion is that a tentative answer can be somehow derived from the experimental data provided by Benavides-Piccione et al 2020. Human Pyramidal cells are packed in a thick layer preserving the morphology, not exactly the size, of the rodent pyramidal cells. This implies that i) pyramidal

apical and basal dendrites are sparsely organized within the pyramidal layer ii) inhibitory neurons and neurites should be reorganized to quench the activity of excitatory neurons in a way that it is actually unknown. In this work we hypothesize that human interneurons, as well as Pyramidal cells, maintain their morphology features adapting the size and sparsely populating the different layers in compliance with an inhomogeneous cellular density distribution. Starting from the experimental data reported in Montero-Cresco et al 2021 (see fig.7 on the primates CA1 cellular layers organization) only deep pyramidal cells project their basal dendrites to the SO and only superficial pyramidal cells project their apical dendrites to the SR and SLM. Thus, we hypothesize that dedicated interneuron classes of interneurons populating the SO or the SR will target more specifically pyramidal apical or basal tufts, while what we called perisomatic interneurons inhibit the pyramidal cell population acting generically on perisomatic dendrites (basal or apical). Following this picture, we can conclude that the choice of modelling 7 different classes of interneurons provides the basis for translating knew knowledge into the model, as it will be available, starting from the plausible assumption that human interneurons should have different morphologies to provide dedicated inhibition.

I have other minor comments.

4. In line 58, you mentioned that CA1 is half of the entire hippocampus. Did you mean in terms of volume? Do you consider the hippocampus as CA1-4 or as including also DG?

I do not see any mention of that in the text

. To be consistent, it would be useful to mention in the paper the volumes of the different layers and the total volume of CA1, and, since you mentioned it in the abstract, also the total volume of hippocampus.

As correctly suggested by the reviewer, we have added to the main text (Image Labeling) the volumes of the different layers that we considered (CA1-CA4) as they have been as estimated by counting the labeled voxels of the 20 micron resolution image

Sub: 289.2668694563181 mm³

CA1: 547.1450172208279 mm³

CA2: 41.60734025896818 mm³

CA3: 55.64641751322779 mm³

CA4: 109.99470129350084 mm³

5. In lines 135-138, I understood that the cell positioning was generated by using information on synaptic connection. Is it correct? If yes, how?

We thank the reviewer for spotting a mistake in the text. The steps we reported are relative to the entire scaffolding procedure. The text has now been emended and the new figure 1 summarizes the entire procedure. Moreover a Table with synaptic connectivity has been added to the Supplemental material (Table SM-2)

6. This is related to point 1.3. The cell counts could have been stated more directly. You took the number of CA1 PC from Cobb et al 2013 (4,836,111) and rounded to 4.8 M. To me, it is important to mention the original number and subsequent operations.

We have now specified in the text (Neuronal Placement) that the original number of CA1 PC was taken from Cobb and that we rounded the number to the closest integer (4.8 M).

7. In lines 221-2, I read that pyramidal cells are uniformly distributed in stratum pyramidale in rodents. Indeed, also in rodents, the pyramidal cells tend to accumulate towards the SR at least in the ventral hippocampus. Check the studies about superficial and deep pyramidal cells.

Good point. Rodent literature [Soltesz et al 2018, Valero et al 2015] reports a division between deep and superficial pyramidal cells. Our intention was to spot that the majority of pyramidal cells in rodents are aligned within a thin SP layer. The sentence has now been rephrased and commented in the discussion

8. In lines 235-6, it lacks of a reference for the orientation of the axon of pyramidal cells

A few sentences have been added to the revision to consider this point. According to rodent experimental data reported in the public repository MouseLight (<http://ml-neuronbrowser.janelia.org/>) pyramidal axons emerge from cell bodies and project their extension mainly into the SO. Moreover, PC preferentially orient their axons toward the Subiculum with little divergence to the CA3. We hypothesized that human pyramidal axons also maintain a similar geometrical feature.

9. In line 297-8, some numbers are lacking. What is the connection probability from rodent? What is the scale factor?

The synaptic probability has been taken directly from the rodent database Hippocampome. We now provide in the SM a Table summarizing synaptic probabilities for all the possible connection pairs.

10. In figure SM-2, I assume you did not have human-specific information on Perforant pathway associated (PPA) cells and you used data from rodent. In that case, the PPA cells should have the axon targeting SLM since they are associated with the perforant pathway that passes in the SLM. On the contrary, the figure SM-2 shows that the axon is centered in SR.

The reviewer is correct, specific data on human Perforant Pathway associated cells are missing. We thank the reviewer for highlighting this issue. The PPA-like geometries have been drawn erroneously: in the model PPA-soma is correctly placed in the superficial part of the SR and axons project their branching into the SLM as it was already specified in the Methods section. We have now corrected the Figure SM-2.

11. While I appreciate that the simulation is only a test to quantify the resources needed, it is not clear why you excluded interneurons. In theory, given the proportion of interneurons, their inclusion should impact only slightly the computational cost. On the other hand, if you exclude them, you cannot claim you have simulated the entire system. The simulation of the entire network will also show that everything works as expected.

We have presented results of the simulation of the purely excitatory network in Fig. 7 of the main text and of the full network in the supplementary materials Fig SM5-6 and we have also uploaded 2 movies of the simulations.

12. You mentioned that files will be available in EBRAINS. Is it possible to get access to complete the review process?

We have preliminary shared the codes on Code Ocean, to reproduce the pipeline for the review process. We hope that the reviewer will understand that we cannot share the actual database of cell positions and connectivity, at the review time. However, as usual for our group, all code and the full, unrestricted, datafiles will be promptly available on EBRAINS KG for public download, without embargo, should this manuscript be accepted.

I hope my feedback is useful for you and helps you to improve the manuscript. While I distinguished between major and minor comments, I believe that all of them are accessible to you.

Best Regards,
Armando Romani

Reviewer 3#

My understanding is that the main result of this project is a structural network/connectivity model of a human right hippocampal CA1 subregion, at the cellular level, based on the analysis of microscopic images. This model (consisting of 3D soma coordinates and connectivity files) will be provided via EBRAINS Knowledge Graph.

The authors may want to clearly define what is that they actually provide in this work (ie, what's the main tangible output?), and to use the term consistently throughout the paper. In scientific writing there is no need to use multiple synonyms to describe the same thing. Within the first 5 pages the following concepts are used to describe what this work is about: model, computational framework, computational pipeline, software and data, a computational tool, a (computational) method. While all of these terms may oft be used interchangeably, they are not.

We apologize with the reviewer for the misunderstanding in the use of different terms. We have now rephrased in the text that in this work we provide: 1) the 3D soma positions of a human right CA1 hippocampus, 2) the connectivity matrix to be used by the Neuroscience community to perform large-scale single point neuron simulations, 3) the algorithm that has been developed to generate the full CA1 scaffold from high-resolution images. Finally, we had mistakenly named the conversion of the algorithm into a series of codes with different terms. Now we refer to it as a computational method throughout the text.

I'm glad I had the opportunity to look at the code used to produce scaffold model. However, it

remains unclear whether the authors will share the code with the community at a later stage. The code is poorly documented and exhibits numerous instances of unexplained hardcoded values. In its current form, their code would be really hard for someone else to re-use it with a different input dataset for instance. All this leads me to think that the assertion “This unique reconstruction, together with the computational method developed for its generation, provide a resource for the implementation of large-scale computational models of the human hippocampus and, on a broader scale, of the human brain.”, is a bit of an overclaim.

We have intentionally shared an uncommented demo version of codes and functions on Code Ocean, specifically to allow the reviewers to have a glimpse at the pipeline. We hope that the reviewer could understand that we could not share a fully commented and complete workflow at that time. We have now uploaded a commented version of codes and functions in the Code Ocean environment. Moreover, as usual for our group, a fully functional and commented code, together with the complete, unrestricted, datafiles will be promptly available for public download on EBRAINS KG and ModelDB, without embargo, should this manuscript be accepted.

I agree that the reconstructed network will be indeed a very valuable resource, but not the computational method, unless more the authors elaborate more on some choices of parameters in every step of the workflow (ie, how much their connectivity matrix changes when adjusting certain parameters in their multiple algorithms? limitations of each step?). There are no discussions on the limitations of their (computational) methods, though to they do discuss limitations of their assumptions about using morphoanatomical features observed in rodents.

We thank the reviewer for highlighting this issue. In response also to a similar point raised by another reviewer, we have tested the effect of adopting different axonal or dendritic sizes in determining network connectivity properties (indegree, outdegree and connection lengths). Fig.SM 4 shows the results of changing pyramidal cells morphological parameters. Briefly, the connectivity algorithm has been used to calculate 9 different network instances on a subpopulation of 200K pyramidal cells randomly selected from the full model. Parameter space has been explored by changing axonal radius (0.5, 1 and 2 times the original value) and dendritic parameters (0.5, 1 and 2 times the original values of cone radius and heights).

A section outlining the limitations of the method has been added to the discussion.

The authors could also improve their code --including the image segmentation functionality that was missing in the code share with the reviewers -- so it can be truly re-used by the community,

and hopefully tested on a different dataset. The authors have clearly done a lot of work and it'd be nice if others could more easily benefit from it.

An updated version of the code including image labeling and image segmentation has been uploaded on the Ocean Code environment. The functions, which include are now commented and explained.

Abstract:

Line 57: "We have benchmarked the method to CA1 region of a right human hippocampus" There's a word missing. Maybe 'reconstruct', 'generate'?. Please check.

done

Line 60: "probability cloud distributions"

Consider rewording "probability cloud distributions" to "axonal and dendritic probability density functions", or "axonal and dendritic probability densities", or if you really want to use the concept of clouds, perhaps use "axonal and dendritic probability clouds".

Without the contextual words "axonal" and/or "dendritic", the concept of "probability cloud distributions" may sound not only ambiguous in the abstract, but also slightly nonsensical. It's unfortunate that Nolte and colleagues (ref 29) decided to use "cloud-based model" instead of "probabilistic model", though perhaps understandable because it's akin to the electron cloud model (ie, a probabilistic model as opposed to the deterministic atomic orbital model). Also note that Nolte et al never referred to their model as "probability cloud distributions".

As suggested by the reviewer, the "probability cloud distribution" has been substituted by "axonal and dendritic probability density functions". Our intention was to adopt a terminology that could be close to the one adopted by Nolte et al.

Line 63: Consider adding the link to EBRAINS Knowledge Graph. Could be useful for readers.

done

Introduction:

Line 71: Add the following references to the list of tech endeavours: Human Neocortical Neurosolver (<https://hnn.brown.edu/>).

<https://elifesciences.org/articles/51214.pdf>

OpenWorm

(<https://openworm.org/>) <https://royalsocietypublishing.org/doi/10.1098/rstb.2017.0382>

References have been added

Line 79: Add “the” before “cellular level”.

done

Line 86: Maybe define CA and DG the first time you introduce the terms (ie, “dentate gyrus (DG)”)

done

Line 89: The mention of mean-fields models deserves at least a few references. See for instance:

- <https://www.jneurosci.org/content/26/4/1314>

- <https://link.springer.com/article/10.1007/s10548-021-00842-4>

- <https://www.frontiersin.org/articles/10.3389/neuro.10.001.2009/full>

- <https://www.pnas.org/doi/10.1073/pnas.1012168108>

- <https://journals.plos.org/ploscompbiol/article?id=10.1371/journal.pcbi.1007822>

The list above is by no means exhaustive. Just a few examples.

We have added a few references in the text

Line 92: “With respect to connectivity, despite several technological advancements,”
Such as? Please enumerate at least a few of these technological advancements you mention and add the corresponding citations.

A few methods to explore neuronal connectivity have been added

Lines 95-96: “Alternatively, high resolution analyses can be obtained at nanoscopic structure in a narrow field view, e.g., with electronic microscopy”

There’s something odd with this sentence. Do you mean that high resolution analyses enable the study of nanoscopic structures in a narrow field view? Or that high resolution analyses are done at a nanoscopic scale? Please clarify.

High resolution techniques in principle allows to obtain very detailed information even at nanoscopic scale but in very limited field of views preventing the observation of large brain areas. We have now hopefully better clarified it in the text.

Line 105: “unpractical”. Replace with “impractical”.

Done

Line 112: “neuritic”

Is that the correct term? I associate this term to neurodegenerative diseases. Please check.

The term neuritic has been removed.

Results:

Line 140: Missing space between “Figure” and “1”

Done

Line 149: maybe introduce a subtitle like “data/dataset and image labelling”, though the authors mention hippocampal segmentation rather than labelling.

Done

Line 158-161: It’s odd to have something in future tense in Results. This sentence is better suited in the Discussion/Future work, not Results. Also you’d be better presenting Fig 1B in the

Introduction, and Figure SM-1 here in Results. Indeed, if the workflow used to produce the network model of CA1 is a Result in itself, then having Fig SM-1 would make the whole Results section much more clear. Current Fig 1B is not a result because if I understood correctly, co-simulations have not been run in this work.

Correct. We have moved the Fig. SM-1 to the results section and is the new Figure 1. The old Fig 1A has been incorporated into the old Figure 2.

Line 194: “remaining (0.48M)”
For completeness reword to “remaining 10% (0.48M)”

done

Lines 199-200; 221-225: Spell out acronyms like OLM and SP, SR, SO in the main text, at the very least the first time they are used.

done

Line 200: “homogeneously distributed”
Perhaps you mean “uniformly distributed”? Or perhaps you refer to isotropically vs anisotropically distributed? Please clarify.

done

Lines 227: “average of the first 4 voxels”
Mean, median or mode?

Mean, it is now stated in the text

Line 233: Replace “with a notable difference.” with “with a notable difference: In the human ...”

The sentence has been rephrased

Figure 6. Indegree and Outdegree.

Lines 303 and 306: “enlarges the curve” perhaps replace with “increases and shifts the peak”. Also mention by how much the peak increases and by how much it is shifted. Express these changes either as a %, or absolute values, or both.

The figure 6 has been further commented with quantitative variations

Line 305: “an average of incoming input of about 3,500 units”. Does this value correspond to the median?
Please clarify.

We have rephrased this sentence. The values are referred to the peak of the distribution

Line 306: Panel C. It'd be great if you could add the histogram/probability density of empirical connection lengths that your histogram is similar to.

Unfortunately, experimental data specifically on the human hippocampus are not available; we have therefore clarified in the text that the similarity of our connection length distribution refers to a general property of neuronal circuits, not only hippocampus. We have now rephrased the sentence.

Line 316: “a shape consistent with experimental data.”
What experimental datasets in particular do you refer to? Refs 45 and 46? Add corresponding refs at the end of this sentence to disambiguate.

done

Lines 317-318: “A similar profile was conserved when inhibitory connections were included in the distribution. The distance distribution also showed a shape similar to that observed experimentally.”
How did you quantitatively assess the similarity between any pair of distributions?

Similarities among distribution have been estimated through the Kullback-Leibler divergence method.

Lines 378-379: “suggesting an anisotropic cells’ spatial density”. I’m not sure I understand this

part of the sentence. Do you mean that your results suggest 'cell density is spatially anisotropic'? Or something else. Try to reword if possible.

done

Discussion:

Lines 362-363: "The idea of developing computational models of the activity of entire brain regions at single neuron resolution has been caressed for several years,". Replace "caressed" with "touched upon" or "discussed" or "entertained".

done

Line 379: "the homogeneous distribution of indegree ..."

Maybe qualify as follows: "the spatially homogeneous distribution of indegree ..."

done

Lines 382-393: I think this paragraphs needs more work. I'm not sure what the message the authors want to convey. I'm particularly confused about "The analysis has been performed on the brain of a healthy 63-year-old human subject," followed by the sentence "that sample was taken from a 99-year-old subject suffering from cognitive decline with the consequent alterations of neuronal structures." Were two datasets, belonging to two different subjects, used?

The paragraph in the discussion points out that another database on human hippocampus exists, and it refers to an EBRAINS-KG repository of a 99-yo woman human hippocampus obtained through light-sheet microscopy. We discuss about differences in the two datasets and on our choice to use the BigBrain dataset instead.

Methods:

Line 482: Specify which version of Matlab was used. Any dependency of a program, including the language itself can change its behaviour from one version to the other.

We have used the v2019b of Matlab and it is now stated in the text. There are no dependencies on previous or newer versions.

Lines 501-502: “The algorithm implemented morphological operation using the built-in 502 function `bwconncomp` and `regionprop2`”

These functions are not used at all in the code provided. Does it mean that what is described in the manuscript was not provided in the code script “neuronal placement”? Or does it mean that the authors ended up doing something different to what is described?

The description sounds like the steps followed to perform image segmentation, so perhaps it'd be better if the text between lines 480-503 was under its own subsection, and then under “neuronal placement” the authors described only the placement procedure. Or, even simpler, update the subsection title to “image segmentation and neuronal placement”.

The image analysis has been performed with a two-steps procedures working separately on labeled and raw images. The first function allowed to generate meshes of hippocampal regions starting from labeled binarized images. This function includes the “`bwconncomp`” and “`regionprops2`” built-in functions. We have provided the reviewer only with the second function which is related to the process of image segmentation to derive neuronal placement. In the methods section we have mistakenly referred to the two built-in functions the procedure of isolating cells somas. We thank the reviewer for spotting this issue.

A “image labeling” section has been added in the results.

Line 535: Mayavi's reference document is <https://hal.archives-ouvertes.fr/hal-00502548>

done

Line 553: “Each voxel sampling was repeated 13 times to account for local density variability.” Why 13 specifically? Could you elaborate on the reasoning for choosing this particular number of repetitions?

The choice of 13 was dictated by geometrical constraints. The idea was to cover the largest volume of CA1 during the density mapping. We have added in the introduction section that the CA1 volume resulting from image analysis is about 547 mm^3 , which can be covered with about 20 repetition of the 25 voxels sampling given the size of a single voxel of $1,000 \times 1,000 \times 1,000 \mu\text{m}^3$. Unfortunately, the CA1 morphology is non-uniform and changes size with a marked decrease from the anterior to the posterior part. We have therefore decided to reduce the number of repetition (the different voxels collected in a certain volume) down to 13 since it was the minimum volume that covered the posterior tip of the CA1. We have now explained it in the methods section

Line 561: “in respect to the three canonical axes”. Replace with “with respect to”.

done

Line 562: “This parametric description allowed to orient axonal”. Replace with “This parametric description allowed us to” or “This parametric description allows/allowed for the (re)orientation of axonal and ...”

done

Line 687: “rescaled of a 1.5 factor.” to “rescaled by a 1.5 factor”

Where does the 1.5 stem from? Can you elaborate how did you arrive to this rescaling factor?

In response to a similar point raised by another reviewer, in the revision we now explain that the scaling factor derives from the comparison of pyramidal cells morphologies in mouse and in human. As it can be evidenced from fig. 6 of Benavides-Piccione 2020 and from data in literature, human pyramidal dendrites are approximately 150% compared to mice. We have extended this factor to all other interneurons that have been modeled starting from data collected from rodents and extended to human

Line 728: “where $0 \leq \theta \leq 2$ ”

I believe it's “where $0 \leq \theta < 2$ ”.

Done

Line 731: as “a” convex hull or “as convex hulls”

Done “convex hulls”

Line 733: “ μm^3 ”

Missing superscript format.

done

Lines 750-751, 756: The logical operator for “AND” is a wedge \wedge , not $\&$ – that’s matlab-specific syntax. Fix or use the word AND to clarify.

done

Line 762: “the numerosity of connections”
Reword to “the number of connections”.

done

Lines 766-767: Almost textual duplication of lines 757-758. Remove.

done

Line 771: “the numerosity of each neuronal class”
Maybe reword as “the number of each neuronal class”, or alternatively you could use “cardinality”, ie, “the cardinality of each neuronal class”.

done

Lines 774-776: It’s difficult to say if the code I looked was the one parallelised, but if it was, there are a few basic things that could be done to speed up the process: (1) write functions rather than scripts. Matlab’s Just-In-Time (JIT) compiler optimises functions, not scripts; (2) heed the advice of Matlab’s linter; and, (3) use a very recent version of Matlab if possible (ie, 2022a) since they have introduced additional optimisations.

We thank the reviewer for the helpful suggestion. As previously stated, the code that has been provided to the reviewer should be used to understand the algorithm and the computational procedures developed to place neurons, generate probability density functions and to connect neurons. The code is based on functions rather than scripts. Moreover, beside a few parallelized computations in the codes, the main parallelization was obtained by splitting the connectivity into chunks of file that could run in parallel on different CPUs of the HPC.

References:

Please fix your references.

- There's a mix of full journal names and abbreviated journal names.
- Duplicated year in some entries.
- Ref 6 points to a conference poster, rather than to the reference paper in Frontiers in Neuroinformatics.
- Inconsistent style to refer to volume and issue numbers.
- Missing authors in some entries.

done

Supplementary Material:

Figure SM-1. Workflow for the generation of human CA1 circuit:

- typo in the caption title: "circuit" replace with "circuit"

done

- "cells coordinated" replace with "cells coordinates"

done

- case: "bigbrain" to "BigBrain"

done

- "somas positions" to simply "soma positions"

done

- "to tun simulation" to "to run simulations" or "to simulate network dynamics".

done

Figure SM-2 Probability clouds of interneurons.
This figure is a diagram correct? It's probably worth mentioning that in the caption title, ie, "Diagram of probability densities of interneurons".

done

Figure SM-3 Indegree and outdegree distribution of the null model.
Plots on the top row are unnecessary. Remove.

done

Decision Letter, first revision:

Dear Dr. Gandolfi,

Thank you for submitting your revised manuscript "Full-scale scaffold model of the human hippocampus CA1 area" (NATCOMPUTSCI-22-0634A). It has now been seen by the original referees and their comments are below. We apologize for the delay in sending a decision to you. Your manuscript was initially sent to 3 reviewers but during this round of review, reviewer #3 was unresponsive. Reviewer #2 has been kind enough to take a look at their reviews and their comments are appended below.

The reviewers find that the paper has improved in revision, and therefore we'll be happy in principle to publish it in Nature Computational Science, pending minor revisions to satisfy the referees' final requests and to comply with our editorial and formatting guidelines.

TRANSPARENT PEER REVIEW

Nature Computational Science offers a transparent peer review option for original research manuscripts. We encourage increased transparency in peer review by publishing the reviewer comments, author rebuttal letters and editorial decision letters if the authors agree. Such peer review material is made available as a supplementary peer review file. **Please state in the cover letter 'I wish to participate in transparent peer review' if you want to opt in, or 'I do not wish to participate in transparent peer review' if you don't.** Failure to state your preference will result in delays in accepting your manuscript for publication.

Please note: we allow redactions to authors' rebuttal and reviewer comments in the interest of

confidentiality. If you are concerned about the release of confidential data, please let us know specifically what information you would like to have removed. Please note that we cannot incorporate redactions for any other reasons. Reviewer names will be published in the peer review files if the reviewer signed the comments to authors, or if reviewers explicitly agree to release their name. For more information, please refer to our [FAQ page](https://www.nature.com/documents/nr-transparent-peer-review.pdf).

Thank you again for your interest in Nature Computational Science Please do not hesitate to contact me if you have any questions.

Sincerely,

Ananya Rastogi, PhD
Associate Editor
Nature Computational Science

ORCID

Reviewer #1 (Remarks to the Author):

The authors have addressed all my queries satisfactorily.

Reviewer #2 (Remarks to the Author):

Dear Authors,

Thanks for having considered all my points. I appreciated the substantial revision you have done and I am satisfied with most of your replies. In few cases, I would ask you for additional clarifications.

I will follow the same numbering as before for clarity.

1.1. As I understood, the number ~150 um came from the analyses of Janalia MouseLight, is it correct?

1.2. I still cannot map your numbers from SM-1 to data in literature or public repositories. This prevents a full reproducibility of your work. Did you download morphologies from neuromorpho or MouseLight and analyze them? Did you also consider values from Bezaire and Soltesz, 2013 or other papers. As said, you mentioned Pelkey et al. 2017, but I could not find the parameters there. It would be useful if you could be more specific. For example, neurite extend were calculated based on morphologies available in neuromorpho or MuouseLight. Axonal extents are taken also from table 5 of Bezaire and Soltesz, 2013. If the sources are multiple, it would be useful to list them as an extra column in Table SM-1.

Related to the previous point, also the distributions of soma, axon and dendrites are a bit strange to me and not well justified.

Perisomatic cells, OLM cells, and Trilaminar cells look good, but not the others.

Ivy cells -> axon and dendrites seem too big, they cover the four layers

SCA cells -> axon is supposed to be mainly in SR, while it is centered in SP

PPA cells -> a classical PPA cell has soma, axon, and dendrites in SLM

https://hippocampome.org/php/neuron_page.php?id=4006

NGF cells -> According to hippocampome the axon is confined in SLM.

https://hippocampome.org/php/neuron_page.php?id=4012

You often cite Pelkey et al. 2017, why did you not use the distribution of soma and neurites shown in their figure 1? In some cases, their distributions differ from yours.

2. In your reply, you wrote interneuron-specific, but I do not think you have it in your model. On the other hand, you forgot OLM, which you have. I think it was a typo.

Apart that, I am not fully convinced by your answer. Without a precise rationale, the choice seems rather arbitrary.

You cited several papers, but they proposed different classes than yours. For example, Pelkey et al. 2017 proposed 15 classes (note that the distributions of soma, axons and dendrites are sometime in conflict with your figure SM-2). Davila et al 2021 proposed "8 GABAergic neuron subpopulations". The justification of different morphological classes is not robust. For example, why did you exclude interneuron-specific?

Maybe the choice was dictated by the data availability. For example, you may say that you chose the 8 classes of cells because you found sufficient examples in neuromorpho or MouseLight, from which you extracted the morphometrics.

For me, all the other comments have been sufficiently addressed.

Best Regards,
Armando Romani

Reviewer #2's comments on the code:

The code run and produces results. Anyway, it is not totally clear what the code is doing. Here, some comments.

1) the script "run" calls Serial_PC_dendrites, but what Serial_PC_dendrites does should be explain better.

1.1) Does it synthesise PC dendrites?

1.2) What are these chunks of 50k? Segments or sections?

1.3) Why chunks of 50k? Because are you printing the results in chunks of 50k?

1.4) Do the PC have always the same number of segment/section? A neuron can be described in one or two files, correct?

1.5) The result is not in a standard format for neuronal morphologies. Could you explain the format in the script?

1.6) It is difficult to edit the script to test a smaller run (see comment 3).

2) in general the code contains very few comments and it is pretty hard to follow and edit it

3) I suspect that not all the files are used in the test run. I would suggest to expand the test run by including all the different steps to build the model together with an explanation.

To avoid that reviewers accidentally run a big stack, you can comment most of them and maybe leave only one uncommented.

4) it would be useful to see a simulation of a subset of neurons.

Reviewer #2's comments on Reviewer #3's review:

I went through reviewer 3's comments and I verified the authors addressed all the points with very few exceptions which I am going to list below.

The code is poorly documented and exhibits numerous instances of unexplained hardcoded values...

I understand the authors do not want to be scooped, but a code poorly documented is also difficult to review.

Line 79: Add "the" before "cellular level".

The authors forgot to add "the".

Author Rebuttal, second revision:

Dear Editor,

Thank you again for giving us the opportunity to further revise the manuscript.

All the points raised have been considered and the text has been amended accordingly.

Below we attach a point-by-point response to reviewers.

We hope the manuscript is now in the form requested for publication in Nature
Computational Science

On behalf of the Authors

Best regards

Daniela Gandolfi

Reviewer #1 (Remarks to the Author):

The authors have addressed all my queries satisfactorily.

Reviewer #2 (Remarks to the Author):

Dear Authors,

Thanks for having considered all my points. I appreciated the substantial revision you have done and I am satisfied with most of your replies. In few cases, I would ask you for additional clarifications.

I will follow the same numbering as before for clarity.

1.1. As I understood, the number ~150 um came from the analyses of Janalia MouseLight, is it correct?

Correct, this value has been estimated by analyzing the Janelia Mouselight database on mouse Pyramidal Cells, rescaled by a 1.5 factor as reported in the main text.

1.2. I still cannot map your numbers from SM-1 to data in literature or public repositories. This prevents a full reproducibility of your work. Did you download morphologies from neuromorpho or MouseLight and analyze them? Did you also consider values from Bezaire and Soltesz, 2013 or other papers. As said, you mentioned Pelkey et al. 2017, but I could not find the parameters there. It would be useful if you could be more specific. For example, neurite extend were calculated based on morphologies available in neuromorpho or MuouseLight. Axonal extents are taken also from table 5 of Bezaire and Soltesz, 2013. If the sources are multiple, it would be useful to list them as an extra column in Table SM-1.

As correctly observed by the reviewer, the parameters adopted to generate neuronal morphologies have been obtained by integrating data collected from the analysis of mouse CA1 morphologies public repositories (Neuromorpho.org and Mouselight) with published data either in the form of tables (Bezaire and Soltesz 2013) or images showing morphological reconstructions (Pelkey et al 2017, Booker et al 2016, Price et al 2005). Parameters obtained from mouse have been rescaled by a 1.5 factor obtained from the comparison between human and mouse CA1 pyramidal neurons. As suggested by the reviewer, to better clarify the multisource origin of the parameters we have added an additional column to table SM-1 with references.

Related to the previous point, also the distributions of soma, axon and dendrites are a bit strange to me and not well justified.

We thank the reviewer for raising this point, because it allowed us to notice mere graphical representation errors in Fig. SM-2. We thus decided to replace the figure SM-2 with a new graphical representation displaying interneuron soma locations and probability density functions drawn over an inset of one of the original BigBrain Nissl stained image. This new version of the figure provides a reference frame for the size of each cloud in relation to the actual scale of the human CA1 derived from data.

Perisomatic cells, OLM cells, and Trilaminar cells look good, but not the others.

Below we give more information for each type of cell. The relevant text in the revision has been expanded.

Ivy cells -> axon and dendrites seem too big, they cover the four layers

In Pelkey et al 2017, Ivy cells are reported to be mainly distributed within the SP and SR but they also populate SO. Moreover, Pelkey et al 2017 report a prevalence of axonal and dendritic extension of Ivy cells within SR, SP and SO. However, it is reported a protrusion within the SLM for both axons and dendrites of Ivy cell cit. "located in superficial SR has been found to extend significant dendritic and axonal process into SLM"

SCA cells -> axon is supposed to be mainly in SR, while it is centered in SP

As shown in the figure below (Fig 1) taken from Pelkey et al 2017, we have modeled mouse SCA cells according to Pelkey et al 2017, in which SCA axons are reported to cit. “co-align with glutamatergic inputs from CA3, ramifying dominantly within SR and to a lesser extent in SO to primarily target the oblique and basal dendrites of pyramidal cells”. We have then extended the mouse representation to human CA1.

We have also attached (Fig. 1) the original Pelkey et al 2017 Fig 3B reproducing a cell reconstruction of a mouse CA1 SCA cell.

[REDACTED]

Fig 1. Reproduction of Fig. 3B of Pelkey et al 2017 showing the reconstruction of a mouse CA1 SCA cell

PPA cells -> a classical PPA cell has soma, axon, and dendrites in SLM

https://hippocampome.org/php/neuron_page.php?id=4006

We have modeled the PPA cells according to Pelkey et al 2017, reporting that PPA soma and axons are predominantly within the SLM or close to the SLM/SR border. Regarding PPA dendrites, we have also integrated data from Booker et al 2017 (which is now included into the references of the main file). As it can be inferred from Fig. 2 below (taken from Booker et al 2017), CA1 mouse PPA dendrites project ramifications into the SR reaching the SP.

[REDACTED]

Fig 2. Reproduction of Fig. 4C of Booker et al 2017 showing the reconstruction of a mouse CA1 PPA cell

NGF cells -> According to hippocampome the axon is confined in SLM.

https://hippocampome.org/php/neuron_page.php?id=4012

As correctly observed by the reviewer, in Pelkey et al 2017 and in Price et al 2005, CA1 mouse NGF cells are confined within the SLM cit. “with a minority of cells residing at the SLM/SR border and superficial

SR. NGF axons and dendrites are confined within SLM with a limited projection to SR". The modeling strategy for CA1 NGF cells respected these morphological constraints.

You often cite Pelkey et al. 2017, why did you not use the distribution of soma and neurites shown in their figure 1? In some cases, their distributions differ from yours.

Please see the replies above.

2. In your reply, you wrote interneuron-specific, but I do not think you have it in your model. On the other hand, you forgot OLM, which you have. I think it was a typo. Apart that, I am not fully convinced by your answer. Without a precise rationale, the choice seems rather arbitrary.

Correct, interneuron specific cells have not been included in our model. There was a typo in the response.

You cited several papers, but they proposed different classes than yours. For example, Pelkey et al. 2017 proposed 15 classes (note that the distributions of soma, axons and dendrites are sometime in conflict with your figure SM-2). Davila et al 2021 proposed "8 GABAergic neuron subpopulations".

The justification of different morphological classes is not robust. For example, why did you exclude interneuron-specific?

Maybe the choice was dictated by the data availability. For example, you may say that you chose the 8 classes of cells because you found sufficient examples in neuromorpho or MouseLight, from which you extracted the morphometrics.

We have included in the model only interneuron classes that are consistently reported in multiple papers. For instance, interneuron-specific are reported in Pelkey et al 2017 but not in Bezaire and Soltesz 2013. As noted by the reviewer, our choice was dictated by data availability in public repositories (neuromorpho and mouselight) and published papers, which were used to calculate morphological parameters. We have added a sentence in the methods section of the main text.

For me, all the other comments have been sufficiently addressed.

Best Regards,
Armando Romani

Final Decision Letter:

Dear Dr Gandolfi,

We are pleased to inform you that your Resource "Full-scale scaffold model of the human hippocampus CA1 area" has now been accepted for publication in Nature Computational Science.

Once your manuscript is typeset, you will receive an email with a link to choose the appropriate publishing options for your paper and our Author Services team will be in touch regarding any additional information that may be required.

Please note that Nature Computational Science is a Transformative Journal (TJ). Authors may publish their research with us through the traditional subscription access route or make their paper immediately open access through payment of an article-processing charge (APC). Authors will not be required to make a final decision about access to their article until it has been accepted.

Authors may need to take specific actions to achieve with funder and institutional open access mandates. If your research is supported by a funder that requires immediate open access (e.g. according to Plan S principles then you should select the gold OA route, and we will direct you to the compliant route where possible. For authors selecting the subscription publication route, the journal's standard licensing terms will need to be accepted, including self-archiving policies. Those licensing terms will supersede any other terms that the author or any third party may assert apply to any version of the manuscript.

Acceptance of your manuscript is conditional on all authors' agreement with our publication policies (see <https://www.nature.com/natcomputsci/for-authors>). In particular your manuscript must not be published elsewhere and there must be no announcement of the work to any media outlet until the publication date (the day on which it is uploaded onto our web site).

Before your manuscript is typeset, we will edit the text to ensure it is intelligible to our wide readership and conforms to house style. We look particularly carefully at the titles of all papers to ensure that they are relatively brief and understandable.

Once your manuscript is typeset and you have completed the appropriate grant of rights, you will receive a link to your electronic proof via email with a request to make any corrections within 48 hours. If, when you receive your proof, you cannot meet this deadline, please inform us at rjsproduction@springernature.com immediately.

If you have queries at any point during the production process then please contact the production team at rjsproduction@springernature.com. Once your paper has been scheduled for online publication, the Nature press office will be in touch to confirm the details.

Content is published online weekly on Mondays and Thursdays, and the embargo is set at 16:00 London time (GMT)/11:00 am US Eastern time (EST) on the day of publication. If you need to know the exact publication date or when the news embargo will be lifted, please contact our press office after you have submitted your proof corrections. Now is the time to inform your Public Relations or Press Office about your paper, as they might be interested in promoting its publication. This will allow them time to prepare an accurate and satisfactory press release. Include your manuscript tracking number **<redacted>** and the name of the journal, which they will need when they contact our office.

About one week before your paper is published online, we shall be distributing a press release to news organizations worldwide, which may include details of your work. We are happy for your institution or funding agency to prepare its own press release, but it must mention the embargo date and Nature Computational Science. Our Press Office will contact you closer to the time of publication, but if you or your Press Office have any inquiries in the meantime, please contact press@nature.com.

We welcome the submission of potential cover material (including a short caption of around 40 words) related to your manuscript; suggestions should be sent to Nature Computational Science as electronic files (the image should be 300 dpi at 210 x 297 mm in either TIFF or JPEG format). We also welcome suggestions for the Hero Image, which appears at the top of our home page; these should be 72 dpi at 1400 x 400 pixels in JPEG format. Please note that such pictures should be selected more for their aesthetic appeal than for their scientific content, and that colour images work better than black and white or grayscale images. Please do not try to design a cover with the Nature Computational Science logo etc., and please do not submit composites of images related to your work. I am sure you will understand that we cannot make any promise as to whether any of your suggestions might be selected for the cover of the journal.

Best,
Fernando (on behalf of Ananya Rastogi)

--

Fernando Chirigati, PhD
Chief Editor, Nature Computational Science
Nature Portfolio